# Assessing Multi-Criteria Decision Analysis Models for Predicting Groundwater Quality in a River Basin of South India

**M. Annie Jenifer [1], Madan Kumar Jha [2,*] and Amina Khatun [2]**

1   VIT School of Agricultural Innovations and Advanced Learning (VAIAL), Vellore Institute of Technology (VIT), Vellore 632014, India; anniejenifer@vit.ac.in
2   AgFE Department, Indian Institute of Technology (IIT), Kharagpur 721302, India; aminakhatun9286@gmail.com
*   Correspondence: madan@agfe.iitkgp.ac.in

**Abstract:** India is the largest consumer of groundwater in the world, and it suffers from a groundwater crisis due to the overexploitation of groundwater and the deterioration of its quality at an alarming rate. Rapid urbanization, a growing population, and mismanagement are major driving forces behind these groundwater issues. Thus, increasing problems of water scarcity and water-quality deterioration threaten the sustainability of the water supply. This necessitates the development of novel approaches to assess prevailing groundwater quality scenarios at a large scale, which can help protect this vital freshwater resource from contamination. In this study, for the first time, the effectiveness of three Geographical Information System (GIS)-based Multi-Criteria Decision Analysis (MCDA) models (i.e., 'Unit Weight', 'Rank Sum', and 'Analytic Hierarchy Process') was explored for predicting groundwater quality in a river basin of Southern India. The seasonal concentrations of groundwater quality parameters, viz., $Cl^-$, TDS, TH, $F^-$, $NO_3^-$-N, $Na^+$, $Mg^{2+}$, $Ca^{2+}$, $K^+$, and $SO_4^{2-}$, were considered for generating their thematic layers. Each thematic layer was classified into suitable feature classes based on the WHO guidelines for drinking water. The thematic layers and the feature classes of individual groundwater quality parameters were assigned relative weights according to the theories of the three MCDA models mentioned above. These thematic layers were then aggregated in GIS to develop Groundwater Quality Index (GQI) maps of the study area for pre-monsoon and post-monsoon seasons. Furthermore, the accuracy of the developed GQI maps was validated using relative operating characteristic curves. The results of the validation indicated that the GIS-based Analytic Hierarchy Process (AHP) model outperformed with prediction accuracies of 71.4% in the pre-monsoon season and about 85% in the post-monsoon season. However, the performances of the Unit Weight and Rank Sum models were found to be average with prediction accuracies varying from 68% to 63% and 64% to 68%, respectively. Thus, the GIS-based AHP model can serve as a reliable scientific tool for predicting seasonal groundwater quality at a river basin scale. It can be very helpful to the policymakers for devising viable management strategies for groundwater protection as well as for ensuring a sustainable water supply.

**Keywords:** groundwater quality prediction; GIS modeling; AHP; unit weight model; rank sum model; groundwater contamination

## 1. Introduction

India is the largest consumer of groundwater in the world, abstracting 251 km³ of groundwater per year [1]. The major driving forces behind this huge groundwater abstraction are the alarming rate of population growth, rapid urbanization, and the mismanagement of groundwater resources. In India, the states of Tamil Nadu, Andhra Pradesh, and Rajasthan hold more than half of the threatened groundwater resources in the country [2]. In Tamil Nadu, 37% of blocks fall under 'Overexploited' (stage of groundwater development greater than 100%), while 8% and 15% of blocks fall under 'Critical' (stage of

groundwater development ranges from 90–100%) and 'Semicritical' (stage of groundwater development ranges from 70–90%) stages of groundwater development [3]. Aside from the depletion of groundwater resources, the falling water tables induce deterioration in the quality of groundwater, which is the predominant issue in India in general and Tamil Nadu in particular. Anthropogenic activities such as the indiscriminate dumping of solid wastes, improper disposal of liquid wastes, and the excessive use of chemical fertilizers and pesticides have led to the deterioration of groundwater quality, which threatens the sustainability of the resource. The contamination of the aquifer is an almost irreversible process [4,5] due to the relatively slow movement of water and pollutants in the subsurface environment. Therefore, there is an urgent need to prevent contamination of groundwater since it is more feasible than the remediation of polluted groundwater resources.

Although investigations of groundwater quality have been primarily based upon field and laboratory research, Geographical Information System (GIS)-based techniques are very effective in advancing towards new frontiers in various groundwater related issues [6]. GIS-based techniques have proved to be effective in organizing, quantifying, and interpreting large volumes of spatial data, providing an efficient environment for data management [7]. The GIS-based Water Quality Index (WQI) is a mathematical technique that translates a large amount of water quality information into a single comparable value [8,9]. WQI is an effective tool in assessing the quality of groundwater [10].

Stigter et al. [11] developed a methodology to compute a Groundwater Quality Index (GWQI) and Groundwater Composition Index (GWCI) by integrating the concentration of several groundwater quality parameters to obtain a dimensionless numerical value. The generated GWQI maps revealed that groundwater quality in the unconfined aquifers was extremely low, greatly affecting the potability of groundwater. Babiker et al. [12] proposed a GIS-based GWQI, to analyze the drinking water suitability of groundwater of Nasuno basin in Tochigi Prefecture of Japan. They also analyzed the sensitivity of the groundwater quality parameters to identify the most significant parameter that highly impacts the spatial variability of GWQI. Yidana et al. [13] analyzed the groundwater quality in the Keta basin, Ghana, using multivariate and spatial techniques. A modified WQI was developed for the study area and they found that mineral weathering and seawater intrusion increased the groundwater salinity in this region. Machiwal et al. [7] carried out a GIS-based assessment and characterization of groundwater quality using long-term and multi-site groundwater quality data of Rajasthan in western India. The results indicated that $Ca^{2+}$, $Cl^-$, and pH are the most influential parameters for cost-effective and long-term groundwater quality monitoring in the study area. Another study by Venkatramanan et al. [14] applied the Canadian Council of Ministers for the Environment Water Quality Index (CCME WQI) technique to generate GQI. This study evaluated the potability of groundwater and recommended some methods for efficient management and protection of groundwater resources.

In the recent past, Selvam et al. [15] developed GIS-based water quality indices for assessing the heavy metal contamination in the Tuticorin district of Tamil Nadu, India. The study identified the highly contaminated site and suggested certain remediation measures for the sustainable management of groundwater resources. Shah and Joshi [16] analyzed the groundwater quality of the Sabarmati river basin of Gujarat, India using the Unit Weight model. They found that the downstream end displayed a lower groundwater quality due to the illegal discharge of sewage, industrial effluent, and urban runoff. Rao and Latha [17] analyzed the groundwater from the hard rock terrain of the Eastern Ghats in Southern India and found that most of the samples collected were not suitable for irrigation. The study also found that the groundwater in this region either lacks necessary minerals or exceeds the recommended concentration, leading to human health hazards. Of late, fuzzy logic has gained popularity and has been employed to assess the quality of groundwater [18–20]. Recently, Jha et al. [21] applied fuzzy logic to estimate the GIS-based GQI in the Tiruchirappalli district of Tamil Nadu, India. This study indicates that the fuzzy indices are more reliable and efficient for groundwater quality assessments. In

addition to this, many researchers have worked on water quality indices for the evaluation of groundwater in India in general [22,23] and Tamil Nadu in particular [14,15,24,25]. It is evident from the literature that although several investigations focus on GIS-based groundwater quality indexing, only a few past studies have employed the AHP model and Unit Weight model for the weight assignment of groundwater quality parameters for GQI calculation. However, the use of the Rank Sum model for GQI estimation has never been applied to date. In addition, none of the past studies have compared the efficiency of these three models in GQI computation. Thus, for the first time, the present study focuses on evaluating the comparative performance of three models, i.e., Unit Weight, Rank Sum, and AHP, in assessing the groundwater quality of the Tiruchirappalli District in Tamil Nadu, India. The entire district of Tiruchirappalli is evaluated for the quality of groundwater which has never been attempted before. In addition, this study proposes a novel approach "Area Under the Curve (AUC) of Relative Operating Characteristic (ROC) curves" for validating the results, which has not been attempted so far.

## 2. Materials and Methods

### 2.1. Overview of the Study Area

The study area is the Tiruchirappalli District, commonly known as Trichy, which has a geographical area of 4403.83 km$^2$ and covers the central part of Tamil Nadu, South India (Figure 1). The study area lies within $10°16'$ and $11°22'$ N latitudes and $78°15'$ and $79°16'$ E longitudes, which is a part of the Cauvery River basin. The mean elevation of the study area is 88 m above Mean Sea Level (MSL), with an average annual precipitation of 841.9 mm. A subtropical climate prevails in this region with the temperature ranging from 29.3 °C to 38.5 °C. The entire district is divided into 14 administrative units (blocks): Anthanallur, Lalgudi, Manachchanallur, Manapparai, Manikandam, Marungapuri, Musiri, Pullambadi, Tattayengarpettai, Tiruverumbur, Thottiyam, Thuraiyur, Uppliyapuram, and Vaiyampatti. Agriculture is the predominant land use, covering 54% of the study area, followed by forests, water bodies, settlements, mining, and industries. The study area is underlain by a hard-rock aquifer system with major hydrogeologic formations such as alluvium, sandstone, limestone, charnockite, granite, and gneiss [26]. Weathered granite and gneiss are predominant in the study area which provides better groundwater yield than charnockitic formations.

Groundwater occurs at a 2–20 m depth in unconfined aquifers (weathered rock formations) [27]. The confined aquifer occurs between 20 and 40 m depths, which is usually fractured or weathered rock formations. The lithological investigation of the study area was carried out using the well-log data of 119 sites (Figure 2). To prepare the geologic profiles of the study area, 16 cross-sections were identified as shown in Figure 2. The geologic profiles along the three cross-sections are shown in Figure 3a–c as an example. It is evident from the geologic profiles that the dominant subsurface formations are various types of gneiss (e.g., granitic gneiss and biotite gneiss), which occurs in weathered and fractured conditions at shallower depths, whereas it occurs in jointed and highly jointed conditions at deeper depths. In addition, weathered and fractured charnockite, sand, and clay occur in some portions of the study area.

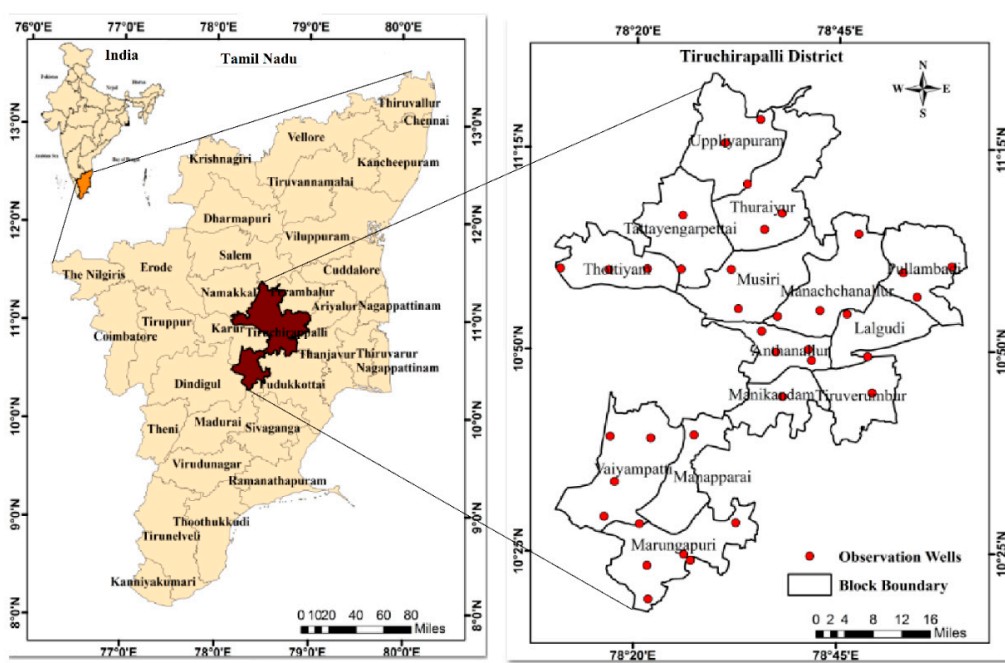

**Figure 1.** Location map of the study area with the locations of observation wells.

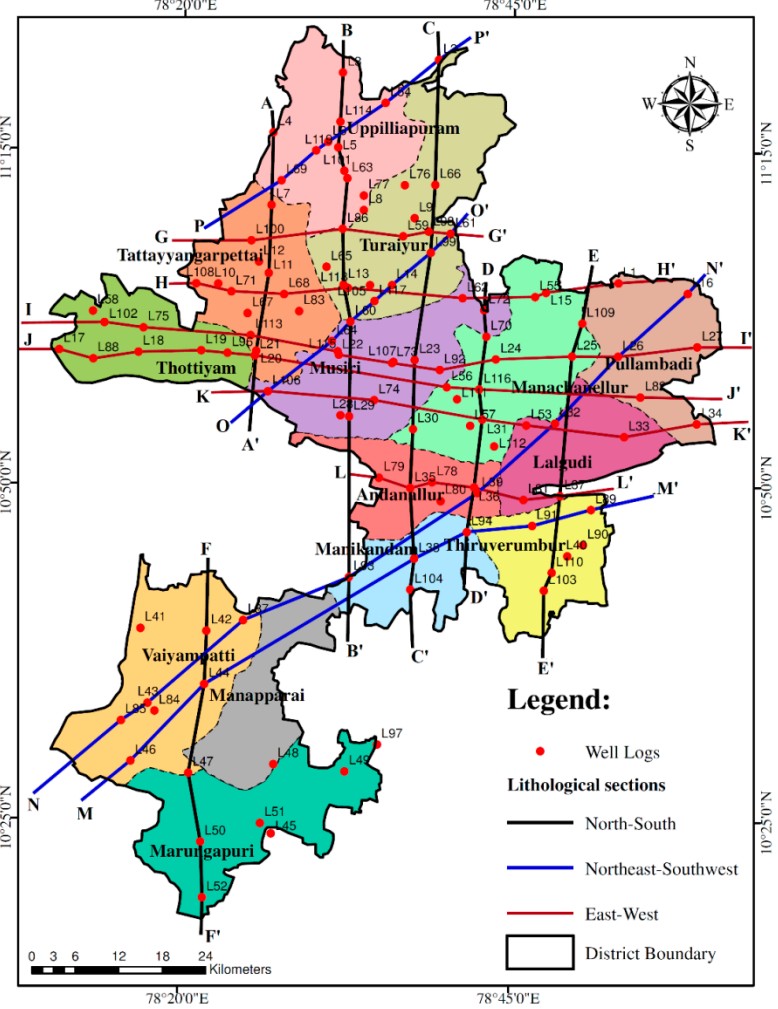

**Figure 2.** Location of well-log sites and 16 cross-sections in the study area.

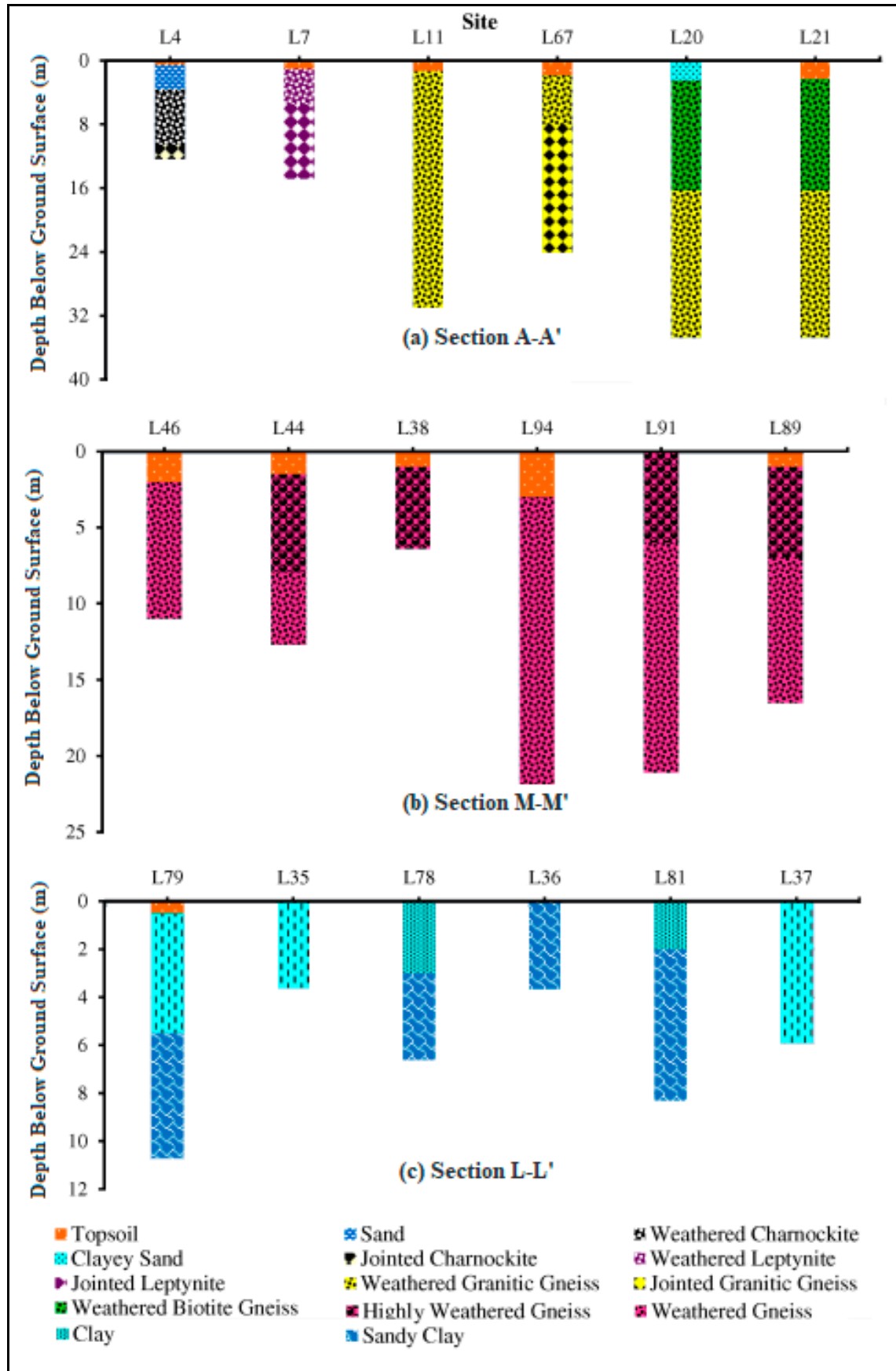

**Figure 3.** (**a–c**). Geologic profiles along: (**a**) North–South (A–A′) Section; (**b**) Southwest–Northeast (M–M′) Section; (**c**) West–East (L–L′) Section.

*2.2. Data Collection and Preliminary Analysis*

The study investigated the groundwater quality data obtained from the groundwater monitoring network (37 observation wells) maintained by the Institute for Water Studies, Chennai, Tamil Nadu. The depth of these observation wells ranged from 92 to 200 m below ground surface. The wells encountered three to five fracture zones with two fracture zones down to 50 m, two fracture zones in the depth of 100–150 m, and one fracture zone deeper than 150 m. These fractured zones yield groundwater through secondary porosity at a rate of 1 to 9 L per second [26]. The pre-monsoon and post-monsoon groundwater quality data of 13 groundwater quality parameters, viz., Chloride ($Cl^-$), Electrical Conductivity (EC), Total Dissolved Solids (TDS), Total Hardness (TH), Fluoride ($F^-$), Nitrate-Nitrogen ($NO_3^-$-N), Sodium ($Na^+$), Magnesium ($Mg^{2+}$), Calcium ($Ca^{2+}$), Potassium ($K^+$), Sulphate ($SO_4^{2-}$), Bicarbonate ($HCO_3^-$) and Carbonate ($CO_3^{2-}$) for a period of 34 years (1981–2014) were obtained from Institute for Water Studies, Chennai, Tamil Nadu.. The parameter EC of groundwater is highly correlated to TDS, whereas $HCO_3^-$ and $CO_3^{2-}$ are highly correlated to TH. Therefore, these parameters are eliminated to avoid duplication in the groundwater quality data. Considering the relative importance of the groundwater quality parameters on drinking water suitability, 10 significant groundwater quality parameters ($Cl^-$, TDS, TH, $F^-$, $NO_3^-$-N, $Na^+$, $Mg^{2+}$, $Ca^{2+}$, $K^+$, and $SO_4^{2-}$) were selected for evaluation in the present study.

A preliminary analysis was carried out using descriptive statistics, namely the mean, standard deviation (SD), coefficient of variation (CV), minimum concentration, and maximum concentration of the salient groundwater quality parameters. Moreover, a trend analysis of the concentrations of selected groundwater quality parameters was performed using the linear regression technique and the significance of the test statistics was checked using a t-test. The correlation analysis was also carried out for the pre-monsoon and post-monsoon seasons to know the relationships between individual pairs of groundwater quality parameters. Finally, the pre-monsoon and post-monsoon groundwater quality data of the year 2014 were used to develop the Groundwater Quality Indices (GQI) for the study area to evaluate the suitability of groundwater for drinking.

*2.3. Assessment of Groundwater Quality Indices*

2.3.1. Generation of Concentration Maps

The point data on the concentration of groundwater quality parameters were interpolated using the Inverse Distance Weighted (IDW) technique in the ArcGIS environment using ArcGIS 10.2. The thematic layers thus generated represent the spatial distribution of the concentrations of groundwater quality parameters during the pre-monsoon and post-monsoon seasons. Furthermore, each thematic layer was classified into suitable feature classes based on the drinking water guidelines of the World Health Organization (WHO). These thematic layers were then used to prepare groundwater quality rating maps. The ratings were calculated using the following equation [28]:

$$q_i = \frac{VA_i - VI_i}{VS_i - VI_i} \times 100 \tag{1}$$

where $q_i$ = the groundwater quality rating for the *i*th parameter, $VA_i$ = the actual concentration of the groundwater quality parameter, and $VI_i$ = the ideal value of the groundwater quality parameter obtained from the standard tables. All the ideal values ($VI_i$) are taken as zero for drinking water except pH [29], and $VS_i$ = the guideline/threshold value for the groundwater quality parameter recommended by WHO. The estimated '$q_i$' values were used to generate the rating maps for individual groundwater quality parameters.

2.3.2. Weight Assignment and Preparation of GQI Map

The prepared rating map for the concentration of each groundwater quality parameter was weighted using three MCDA models: (a) Unit Weight model, (b) Rank Sum model, and (c) Analytic Hierarchy Process (AHP). The methodology adopted for the assignment

of weights to the selected groundwater quality parameters and the development of GQI is explained in the subsequent subsections.

Unit Weight Model

In this model, the weightage factor '$W$' associated with each parameter is estimated using Equation (2):

$$W_i = \frac{K}{S_i} \tag{2}$$

where $W_i$ = the weightage factor for $i$th parameter, $K$ = the proportionality constant, and $S_i$ = the WHO standard value of the $i$th parameter. From Equation (2) it can be seen that the weights assigned to the groundwater quality parameters are inversely proportional to the WHO recommended values for the respective parameter [30].

The proportionality constant '$K$' was obtained using the following equation:

$$K = \frac{1}{\left(\sum\limits_{i=1}^{n} \frac{1}{S_i}\right)} \tag{3}$$

where $S_i$ = the guideline/threshold value of the parameter recommended by WHO.

Finally, the GQI was computed for the pre-monsoon and post-monsoon seasons by integrating the groundwater quality rating ($q_i$) with the weightage factors evaluated for individual groundwater quality parameters using Equation (4):

$$GQI = Anti \log\left[\sum\limits_{i=1}^{n} W_i \log_{10} q_i\right] \tag{4}$$

where '$W_i$' and '$q_i$' were calculated from Equations (1) and (2), respectively.

Rank Sum Model

The Rank Sum model is a direct multi-attribute utility technique for assigning weights to different criteria based on their relative importance. According to this theory, Rank 1 is assigned to the potential parameter which highly influences the drinking water quality of groundwater, whereas Rank 10 is assigned to the less influential parameter. The ranks are then translated into weights by normalizing them to an approximate ratio scale using the weightage factor '$W$' calculated as [31]:

$$W_i = \frac{K - r_i + 1}{\sum\limits_{i=1}^{K} K - r_i + 1} \tag{5}$$

where $W_i$ = the weightage factor of the $i$th parameter, $r_i$ = the rank of the $i$th parameter, and $K$ = the total number of the parameters. The GQI was then computed as the weighted linear combination of weightage factor ($W_i$) and groundwater quality rating ($q_i$) in the GIS environment as follows:

$$GQI = \sum\limits_{i=1}^{n} W_i q_i \tag{6}$$

where '$W_i$' and '$q_i$' were calculated from Equations (5) and (1), respectively.

Analytic Hierarchy Process (AHP) Model

The Analytic Hierarchy Process is an indirect weight assignment technique introduced by T.L. Saaty in 1970. It segregates complex problems into a hierarchy of possible simple alternatives. They are converted to numerical values that are ranked priority-wise on a 9-point scale developed by Saaty [32]. In this study, the concentration maps of the 10 groundwater quality parameters were considered for weight assignment as per Saaty's AHP scale [27].

The pair-wise comparison matrices for each theme (concentration map) and their corresponding feature classes were prepared. The weights assigned were then normalized by the eigenvector technique. In addition, the consistency of the assigned weights was checked using Equations (7) and (8). For the weights to be consistent, the value of CR should be less than 10%, otherwise, the weights need to be modified. Equations (7) and (8) are calculated as follows:

$$\text{Consistency Index (CI)} = \frac{\lambda_{\max} - n}{n - 1} \tag{7}$$

where $\lambda_{\max}$ = the principal eigenvalue, $n$ = the number of themes or feature classes, and:

$$\text{Consistency Ratio (CR)} = \frac{\text{Consistency Index (CI)}}{\text{Random Consistency Index (RI)}} \tag{8}$$

where the value of the Random Consistency Index (RI) was obtained from the standard table given by Saaty [27].

Finally, the normalized weight assigned to the thematic layers and their feature classes were integrated with the help of ArcGIS software using Equation (9) to develop GQI maps for the pre-monsoon and post-monsoon seasons:

$$GQI = \sum_{i=1}^{n} \sum_{j=1}^{m} W_i \times w_j \tag{9}$$

where $W_i$ = the normalized weight of individual themes, $w_j$ = the normalized weight of the features of individual themes, $n$ = the number of themes, and $m$ = the number of feature classes in a theme.

The GQI values thus obtained by the three weighting models for the pre-monsoon and post-monsoon seasons were classified into six groundwater quality classes based on water suitability for drinking (i.e., 'Very Good, 'Good', 'Moderate', 'Poor', 'Very Poor', and 'Unfit') to generate GQI maps of the study area using the 'Natural break' classification technique available in ArcGIS. A flowchart showing the methodology for evaluating groundwater quality using the three GIS-based MCDA models is depicted in Figure 4.

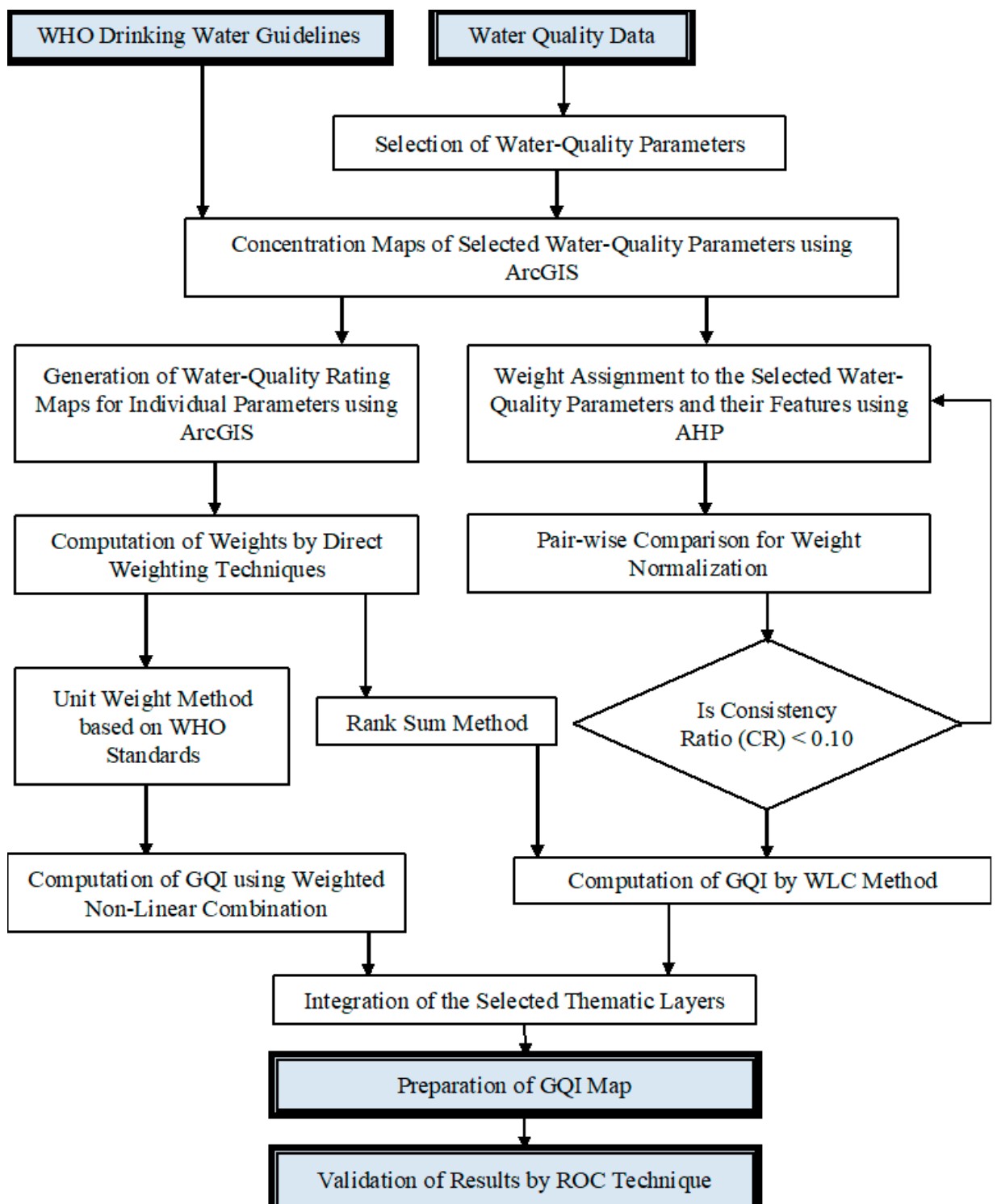

**Figure 4.** Flowchart illustrating procedures for the groundwater quality assessment using Unit Weight, Rank Sum, and AHP models.

### 2.4. Validation of the GQI Maps

In this study, a novel approach the "Relative Operating Characteristics (ROC) curve technique" was adopted for validating the GQI maps obtained by the three GIS-based MCDA models. The ROC curve was originally developed for use in radar signal detection to check the accuracy of output signals [33]. The 'sensitivities' or 'false positive rates' are plotted along the *X*-axis, and the '1-Specificity values' or 'true positive rates' are plotted

along the $Y$-axis at various cut-off values. The "Area Under the Curve (AUC)" is a measure of the ability of the predictive model to predict observed events [33–35]. The accuracy of prediction is evaluated based on the AUC values obtained with the minimum value indicating lesser accuracy, i.e., 0.5–0.6 (Poor), 0.6–0.7 (Average), 0.7–0.8 (Good), 0.8–0.9 (Very Good), and 0.9–1 (Excellent) [36]. In this study, the pre-monsoon and post-monsoon concentration values of groundwater quality parameters were compared with the calculated index value in the GQI maps to obtain a binary value. For example, if the concentration of a parameter from a particular observation well is well within the recommended value and this observation well falls in the 'Very Good', 'Good', or 'Moderate' classes of groundwater quality, then it is provided with a 'true positive rate'. On the other hand, if it falls in the 'Poor', 'Very Poor', or 'Unfit' groundwater quality zone, then it is assigned a 'false positive rate'. Thus, 'true positive' and 'false positive' rates were calculated for 37 observation wells to plot the ROC curves. Three ROC curves were plotted in this study for each of the three models employed for computing the groundwater quality index.

## 3. Results and Discussion

### 3.1. Spatio-Temporal Variation of Groundwater Quality Parameters

The selected ten groundwater quality parameters, viz., $Cl^-$, TDS, TH, $F^-$, $NO_3^-$-N, $Na^+$, $Mg^{2+}$, $Ca^{2+}$, $K^+$, and $SO_4^{2-}$, were analyzed to obtain the spatial and temporal variability over 34-year (1981–2014) period. The descriptive statistics of all the parameters calculated for the pre-monsoon and post-monsoon seasons are presented in Tables A1 and A2, respectively. The results of four blocks, namely, Anthanallur, Manikandam, Tattayengarpettai, and Thuraiyur, are not presented due to the lack of data. It is apparent from Table A1 that during the pre-monsoon season, the concentrations of $Na^+$ (in six blocks), $Ca^{2+}$ (in four blocks), $Mg^{2+}$ (in five blocks), and $SO_4^{2-}$ (in three blocks) exhibit low temporal variability (CV < 10%). Generally, a CV value of less than 10% represents low variability, while the CV values of 10–100% represent moderate variability, and a CV value greater than 100% represents high variability [37] in the groundwater quality parameter under consideration. On the other hand, higher values greater than 100% are found for the $Cl^-$ concentration in the Uppliyapuram block (104.6%), indicating very high temporal variability of $Cl^-$ in groundwater. Apart from this, the CV values of the remaining groundwater quality parameters fall within 10 to 100% indicating their moderate temporal variability. This reveals that the concentrations of these groundwater quality parameters are varying evenly in these blocks of the study area during the 34 years (1981–2014) considered for this analysis. In addition to this, the trend analysis results reveal that the concentrations of TDS (Pullambadi), $Cl^-$ (Lalgudi, Pullambadi, and Thottiyam), TH (Pullambadi), $Na^+$ (Pullambadi and Thottiyam), $Mg^{2+}$ (Pullambadi and Tiruverumbur), and $K^+$ (Lalgudi, Musiri, Thottiyam, and Vaiyampatti) have significantly increased over the past 34 years. In contrast, the concentrations of TDS, $Cl^-$, TH, and $Na^+$ in the Marungapuri block, $NO_3^-$-N in the Musiri and Thottiyam blocks, and $K^+$ in the Tiruverumbur block show a significant declining trend during the pre-monsoon season.

On the other hand, it can be seen from Table A2 that during the post-monsoon season, the concentrations of $Na^+$ (Lalgudi and Thottiyam), $Ca^{2+}$ (Marungapuri, Musiri, and Pullambadi), $Mg^{2+}$ (Manachchanallur, Musiri, Pullambadi, and Vaiyampatti), $K^+$ (Musiri), and $SO_4^{2-}$ (Lalgudi) in the groundwater of these blocks exhibit less temporal variability. The remaining groundwater quality parameters exhibit a moderate temporal variability in the study area. However, it is found that none of the groundwater quality parameters exhibit high temporal variability in the post-monsoon season during the study period. Moreover, the results of the trend analysis revealed that the concentrations of TDS (Thottiyam), $Cl^-$ (Thottiyam), TH (Thottiyam), $Na^+$ (Musiri), $Mg^{2+}$ (Thottiyam and Tiruverumbur), and $K^+$ (Musiri, Pullambadi, and Vaiyampatti) in the groundwater of these blocks have a significant increasing trend. In contrast, a significant decreasing trend is found in the concentrations of TDS (Manapparai), $Cl^-$ (Manapparai), $F^-$ (Tiruverumbur), and $Ca^{2+}$ (Vaiyampatti) in the groundwater of these blocks.

Moreover, the results of the correlation analysis for the pre-monsoon season (Table A3) show high positive correlation (r > 0.5) for TDS and TH with three ions, viz., $Na^+$ (r = 0.86), $Cl^-$ (r = 0.93), and $SO_4^{2-}$ (r = 0.77). This indicates that these ions predominantly contribute to the TDS and TH in the groundwater of the study area. $Cl^-$ is highly associated with $Na^+$ (r = 0.92), while $SO_4^{2-}$ exhibits a close association with $Na^+$ and $Cl^-$ with correlation coefficients of 0.62 and 0.75, respectively. This indicates that these ions may originate from anthropogenic sources such as agricultural fields or domestic wastewaters. There is no significant correlation for potassium, fluoride, and nitrate-nitrogen with the other parameters during the pre-monsoon season. On the other hand, during the post-monsoon season (Table A4) all the ions contribute to the TDS and TH of groundwater except for $K^+$ and $F^-$. Magnesium shows positive correlations with $Ca^{2+}$ (r = 0.85) and $Na^+$ (r = 0.52). $SO_4^{2-}$ exhibits a close association with $Mg^{2+}$, $Na^+$, and $Cl^-$ with correlation coefficients of 0.53, 0.57, and 0.58, respectively. There is no significant correlation of potassium and fluoride with any other groundwater quality parameters considered in this study. In contrast to the pre-monsoon season, $NO_3^-$-N indicates a strong correlation towards six groundwater quality parameters in the order of $Cl^- > TDS > TH > Mg^{2+} > Ca^{2+} > Na^+$. This indicates that the nitrate-nitrogen concentration varies in groundwater along with other parameters due to the infiltration of contaminants during the monsoon rainfall. However, $NO_3^-$-N is poorly correlated with $K^+$ and $F^-$ ions.

### 3.2. Variability of Groundwater Quality in the Study Area

The spatial variability of individual groundwater quality parameters over the study area is shown in Figure 5a–h and Figure 7a–d. The concentration of $Cl^-$ in the pre-monsoon season varies from 35.02 to 1304.97 mg/L [Figure 5a] and 32 to 1346.96 mg/L in the post-monsoon season [Figure 5b]. About 15% and 18% of the study area has $Cl^-$ concentrations exceeding the WHO recommended maximum threshold value (>200 mg/L) for drinking water during the pre-monsoon and post-monsoon seasons, respectively. From Figure 5a,b it can be seen that the concentration of $Cl^-$ ions in the groundwater is within the WHO recommended threshold value of 200 mg/L in the extreme northern part of the study area only. The TDS concentration ranges from 158.005 to 3183.95 mg/L in the pre-monsoon season [Figure 5c] and 190.002 to 3027.93 mg/L in the post-monsoon season [Figure 5d]. In 60% of the study area, the TDS concentration in the groundwater is found to fall within the recommended value (<500 mg/L). Figure 5e,f show that the TH (measured as mg/L of $CaCO_3$) in the groundwater is found to be moderately hard (75–150 mg/L), hard (150–300 mg/L), or very hard (>300 mg/L) in the pre-monsoon season. However, during the post-monsoon season, the TH concentration exceeds 150 mg/L ('hard' or 'very hard') in the entire study area. The $Na^+$ concentration in the groundwater exceeds the threshold value of 200 mg/L in the southern part of the study area during both the pre-monsoon and post-monsoon seasons [Figure 5g,h]. Fluoride toxicity in the groundwater (>1.5 mg/L) can be seen in the Vaiyampatti block during the pre-monsoon season [Figure 6a] and in the Marungapuri and Manikandam blocks during the post-monsoon season [Figure 6b]. The concentrations of $Ca^{2+}$ [Figure 6c,d], $SO_4^{2-}$ [Figure 6e,f], and $Mg^{2+}$ [Figure 6g,h] in the groundwater are within the WHO recommended limits (<70 mg/L for $Ca^{2+}$, <250 mg/L for $SO_4^{2-}$ and <300 mg/L for $Mg^{2+}$) in the entire study area during both seasons. The concentration of $Mg^{2+}$ in the groundwater is found to exceed the 300 mg/L limits in the Vaiyampatti block only during the post-monsoon season. Figure 7a,b depict that the the $NO_3^-$-N concentration exceeds the WHO recommended value of 10 mg/L in almost all the blocks of the study area. In addition to this, the $K^+$ concentration (>10 mg/L) also exceeds the limits in the entire area except for a few portions in the extreme northern and southern parts of the study area [Figure 7c,d].

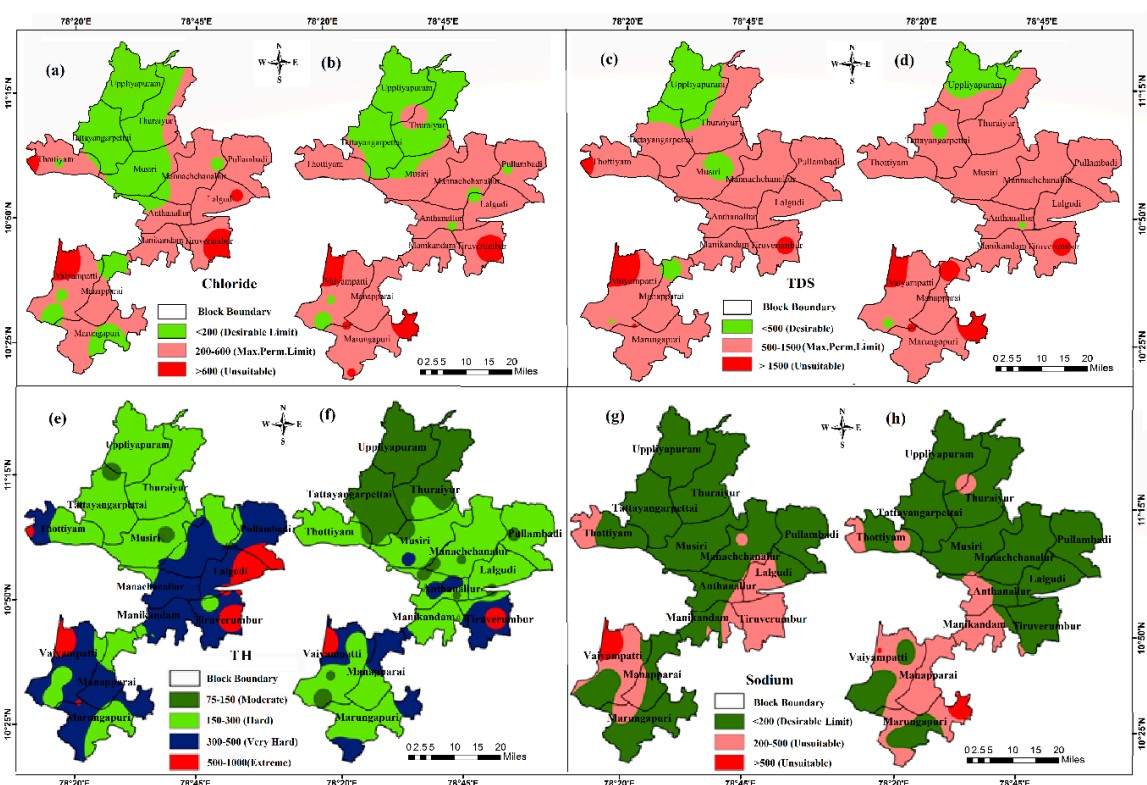

**Figure 5.** (**a**–**h**). Variation of concentration of groundwater quality parameters: (**a**) Cl⁻ in pre-monsoon season; (**b**) Cl⁻ in post-monsoon season; (**c**) TDS in pre-monsoon season; (**d**) TDS in post-monsoon season; (**e**) TH in pre-monsoon season; (**f**) TH in post-monsoon season; (**g**) Na⁺ in pre-monsoon season; (**h**) Na⁺ in post-monsoon season.

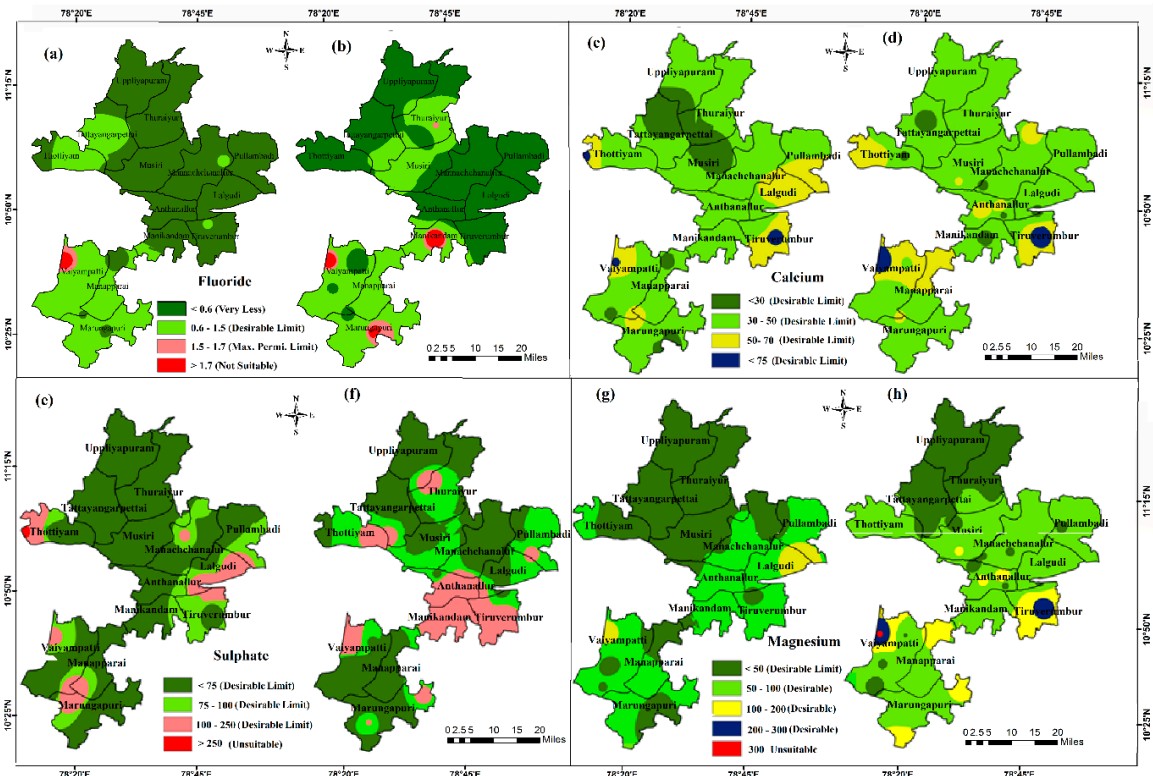

**Figure 6.** (**a**–**h**). Variation of concentration of groundwater quality parameters: (**a**) F⁻ in pre-monsoon season; (**b**) F⁻ in post-monsoon season; (**c**) Ca²⁺ in pre-monsoon season; (**d**) Ca²⁺ in post-monsoon season; (**e**) SO₄²⁻ in pre-monsoon season; (**f**) SO₄²⁻ in post-monsoon season; (**g**) Mg²⁺ in pre-monsoon season; (**h**) Mg²⁺ in post-monsoon season.

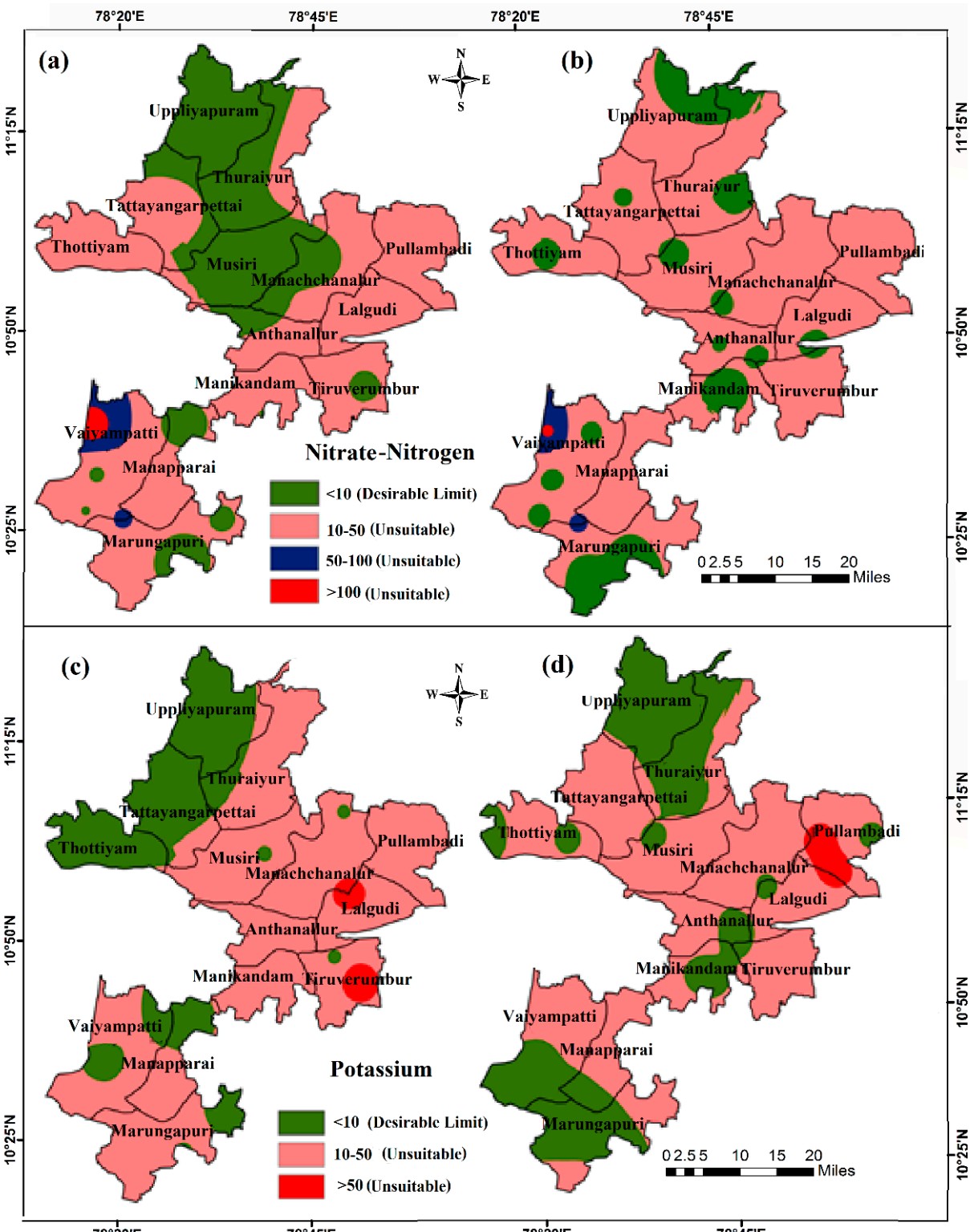

**Figure 7.** (**a**–**d**). Variation of concentration of groundwater quality parameters: (**a**) $NO_3^--N$ in pre-monsoon season; (**b**) $NO_3^--N$ in post-monsoon season; (**c**) $K^+$ in pre-monsoon season; (**d**) $K^+$ in post-monsoon season.

### *3.3. Seasonal Groundwater Quality Index Maps of the Study Area*

3.3.1. GQI Maps Based on the Unit Weight Model

The weights estimated using the Unit Weight model to assess the groundwater quality for drinking in the study area are presented in Table 1. The basic difference of this model

from the AHP model is that it quantifies the concentration maps of the thematic layers by rating them using the WHO guidelines for drinking water. As such, it does not consider the features of individual themes. The groundwater quality rating maps for each of the themes were prepared separately for both seasons using Equation (1). The standard ($S_i$) drinking water guidelines recommended by the WHO were considered as the threshold for the calculation of weights for the groundwater quality parameters. The calculation of weight for $Mg^{2+}$ ion is presented below as an example:

**Table 1.** Weight assessment using **Unit Weight method**.

| Sl. No. | Groundwater Quality Parameters | WHO (2017) Threshold Value or * Guideline Value $S_i$ (mg/L) | $1/S_i$ | K | Weights ($W_i$) |
|---------|-------------------------------|---------------------------------------------------------------|---------|------|-----------------|
| 1 | $Mg^{2+}$ | 300 | 0.0033 | 1.1161 | 0.0037 |
| 2 | * $F^-$ | 1.5 | 0.6667 | 1.1161 | 0.7440 |
| 3 | TDS | 500 | 0.0020 | 1.1161 | 0.0022 |
| 4 | $Ca^{2+}$ | 300 | 0.0033 | 1.1161 | 0.0037 |
| 5 | $Na^+$ | 200 | 0.0050 | 1.1161 | 0.0056 |
| 6 | $SO_4^{2-}$ | 250 | 0.0040 | 1.1161 | 0.0045 |
| 7 | $K^+$ | 10 | 0.1000 | 1.1161 | 0.1116 |
| 8 | $Cl^-$ | 200 | 0.0050 | 1.1161 | 0.0056 |
| 9 | * $NO_3^{-}$-N | 10 | 0.1000 | 1.1161 | 0.1116 |
| 10 | TH | 150 | 0.0067 | 1.1161 | 0.0074 |
| | | | $\sum 1/S_i$ = 0.8960 | | **Total = 1** |

Note: *Threshold Value*: The minimum concentration at which taste or odor sensitivity to a particular constituent in water can be perceived; * *Guideline Value:* A numerical value that represents the concentration of the constituents in water that does not result in any significant risk to human health for life-long consumption.

*WHO recommended value*: The minimum threshold value ($S_i$) of $Mg^{2+}$ recommended by the WHO for drinking purposes is 300 mg/L. Hence, the $1/S_i$ value becomes 0.0033 L/mg. Similarly, the $S_i$ and $1/S_i$ values of all other nine water-quality parameters were calculated and are given in Table 1. The value of proportionality constant '$K$' was obtained from Equation (3) as:

$$K = \frac{1}{\left(\sum\limits_{i=1}^{n} \frac{1}{S_i}\right)} = \frac{1}{0.8960} = 1.1161$$

*Weightage factor*: The weightage factor $W_i$ for $Mg^{2+}$ was calculated using Equation (2) as:

$$W_i = \frac{K}{S_i} = \frac{1.1161}{300} = 0.0037$$

This procedure was repeated for all other groundwater quality parameters (themes) to obtain their relative weights. In this study, the highest weight was obtained for $F^-$ (0.7440) and the lowest weight for TDS (0.0022) using the Unit Weight model. The weightage factors of all the ten themes were then aggregated with the seasonal groundwater quality rating maps in the ArcGIS environment to obtain GQI maps of the study area.

The GQI values estimated based on the Unit Weight model range from 9.83 to 253.35 in the pre-monsoon season [Figure 8a] and 3.98 to 175.06 in the post-monsoon season [Figure 8b]. The estimated GQI values were then categorized to produce GQI maps with six classes of groundwater quality, viz., 'Very Good', 'Good', 'Moderate', 'Poor', 'Very Poor', and 'Unfit'. The areal extent of the groundwater quality classes predicted by the Unit Weight model is presented in Table 2. The pre-monsoon GQI map predicted by the Unit Weight model reveals that a major part of the study area falls in the 'Good' (63%) groundwater quality class followed by the 'Moderate' (20%) and 'Poor' (11%) groundwater

quality classes. Groundwater is found to be 'Very Good' in 4% of the study area, which is confined to the northern part. In contrast, the groundwater quality class is found to be 'Poor' or 'Unfit' in the southwestern parts of the study area [Figure 8a]. As per the Unit Weight-based seasonal GQI maps, Marungapuri, Manapparai, and Vaiyampatti are the three blocks having 'Poor' to 'Very Poor' groundwater quality as far as drinking water suitability is concerned. Rapid industrialization in the central and southern parts of the study area intensifies the concentration of certain groundwater quality parameters in these blocks. In addition, a major portion of the Vaiyampatti block lies under the 'Unfit' groundwater quality category since this block lies in the downstream end of the study area and serves as an outlet for the contaminants. In contrast, the groundwater quality in the post-monsoon season is found to deteriorate with the reduction in the area under the 'Good' (46.3%) groundwater quality class and increase in the area under the 'Poor' groundwater quality (13.3%) class [Figure 8b]. However, during the post-monsoon season, there is a slight increase in the areal extent of 'Very Good' and 'Moderate' groundwater quality classes as compared to the pre-monsoon season. Groundwater is found to be 'Very Poor' or 'Unfit' for human consumption in the Manikandam, Marungapuri, and Vaiyampatti blocks of the study area during the post-monsoon season. This can be attributed to the leaching of phosphate fertilizers from the agricultural fields which are the major sources of $F^-$ in the Vaiyampatti block, while the excessive mining carried out in the Marungapuri and Manikandam blocks increases $F^-$ concentration in the groundwater of these blocks.

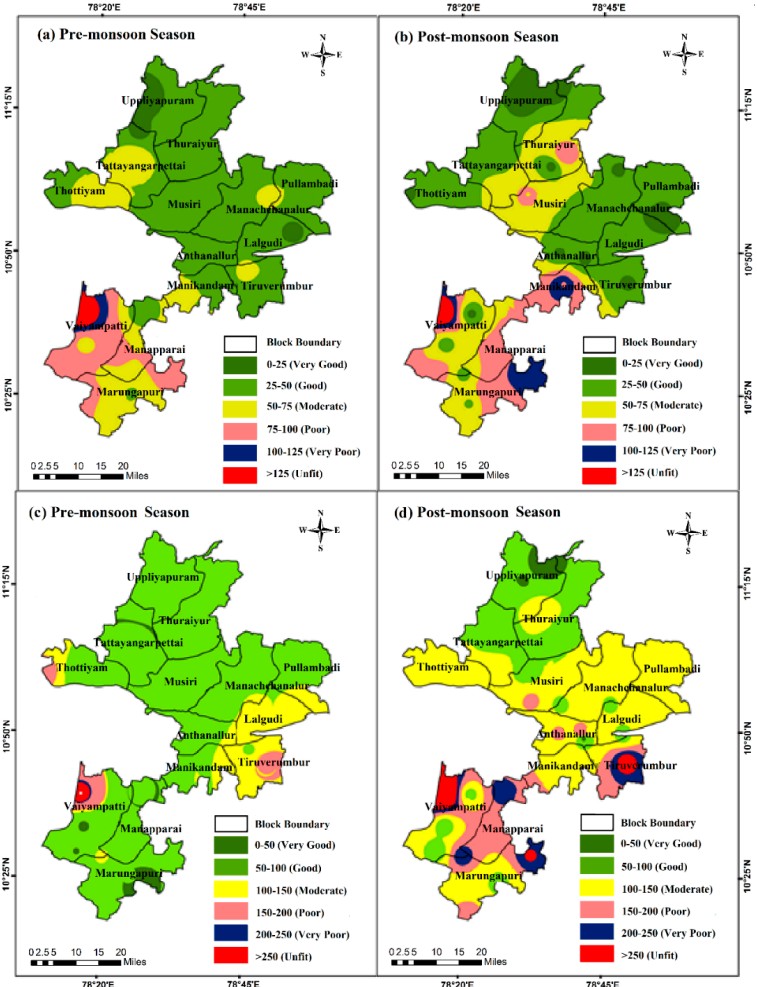

**Figure 8.** (**a–d**). GQI maps of the study area: (**a**) Based on **Unit Weight model** for pre-monsoon season; (**b**) Based on **Unit Weight model** for post-monsoon season; (**c**) Based on **Rank Sum model** for pre-monsoon season; (**d**) Based on **Rank Sum model** for post-monsoon season.



**Table 2.** Areal extent of groundwater quality zones predicted by the **Unit Weight model**.

| Groundwater Quality Zone | GQI Range | Pre-Monsoon Season | | Post-Monsoon Season | |
|---|---|---|---|---|---|
| | | Area (km$^2$) | Area (%) | Area (km$^2$) | Area (%) |
| 1. Very Good | 0–25 | 172.11 | 3.82 | 383.74 | 8.51 |
| 2. Good | 25–50 | 2820.75 | 62.63 | 2086.85 | 46.29 |
| 3. Moderate | 50–75 | 919.98 | 20.43 | 1242.71 | 27.57 |
| 4. Poor | 75–100 | 481.86 | 10.70 | 600.14 | 13.31 |
| 5. Very Poor | 100–125 | 50.69 | 1.13 | 157.12 | 3.49 |
| 6. Unfit | >125 | 58.40 | 1.30 | 37.430 | 0.83 |

### 3.3.2. GQI Maps Based on the Rank Sum Model

Similar to the Unit Weight model, the groundwater quality rating maps were prepared using the Rank Sum model considering the WHO recommended guideline/threshold values. The ranks and weights assigned to all the groundwater quality parameters based on the Rank Sum model are presented in Table 3. Considering the relative importance and toxicity effects on the drinking water quality, $NO_3^-$-N and $F^-$ concentration in groundwater are assigned with the first and second ranks followed by the other parameters, i.e., $Cl^-$, $Na^+$, TH, TDS, $SO_4^{2-}$, $Ca^{2+}$, $K^+$, and $Mg^{2+}$. The steps involved in calculating the weight for $Mg^{2+}$ are shown below as an example.

**Table 3.** Weight assessment using **Rank Sum method**.

| Sl. No. | Groundwater Quality Parameters | Rank ($r_i$) | Total No. of Parameters (K) | $K - r_i + 1$ | Weights ($W_i$) |
|---|---|---|---|---|---|
| 1 | $Mg^{2+}$ | 10 | 10 | 1 | 0.0182 |
| 2 | $F^-$ | 2 | 10 | 9 | 0.1636 |
| 3 | TDS | 6 | 10 | 5 | 0.0909 |
| 4 | $Ca^{2+}$ | 8 | 10 | 3 | 0.0545 |
| 5 | $Na^+$ | 4 | 10 | 7 | 0.1273 |
| 6 | $SO_4^{2-}$ | 7 | 10 | 4 | 0.0727 |
| 7 | $K^+$ | 9 | 10 | 2 | 0.0364 |
| 8 | $Cl^-$ | 3 | 10 | 8 | 0.1455 |
| 9 | $NO_3^-$-N | 1 | 10 | 10 | 0.1818 |
| 10 | TH | 5 | 10 | 6 | 0.1091 |
| | | | | Total = 55 | Total = 1 |

*Weightage factor*: From Table 3, it can be seen that $Mg^{2+}$ is assigned rank 10, i.e., $r_i = 10$. Hence, $K - r_i + 1 = 10 - 10 + 1 = 1$, where $K$ = the total no. of parameters. The weightage factor ($W_i$) for $Mg^{2+}$ was then obtained using Equation (5) as follows:

$$W_i = \frac{K - r_i + 1}{\sum\limits_{j=1}^{K} K - r_j + 1} = \frac{1}{55} = 0.0182$$

Similarly, the weights of the remaining nine parameters were calculated by repeating the above procedure.

Applying weighted linear combination, the seasonal groundwater quality rating maps and weightage factors were integrated to develop GQI maps for the pre-monsoon and post-monsoon seasons based on the Rank Sum model [Figure 8c,d]. The GQI values thus obtained range from 20.22–517.08 in the pre-monsoon season and 21.41–520.44 in the post-monsoon season. It is apparent from Figure 8c,d that the groundwater quality deteriorated in the post-monsoon season compared to the pre-monsoon season. About 79.8% of the study area falls under the 'Good' groundwater quality class in the pre-monsoon season, which was reduced to 24.9% in the post-monsoon season. In addition, there is an increment in the areal extent of the 'Poor', 'Very Poor', and 'Unfit' classes from 3.4%, 0.2%, and 0.5% to 12.3%, 4.7%, and 2.4%, respectively, during the post-monsoon season as compared to the pre-monsoon season (Table 4), which justifies the degrading groundwater quality. However, the areal extent of the 'Moderate' groundwater quality class in the pre-monsoon season, i.e., 14.1% of the study area, increases to 53.9% in the post-monsoon season.

**Table 4.** Areal extent of groundwater quality zones predicted by the **Rank Sum model**.

| Groundwater Quality Zone | GQI Range | Pre-Monsoon Season | | Post-Monsoon Season | |
|---|---|---|---|---|---|
| | | Area (km²) | Area (%) | Area (km²) | Area (%) |
| 1. Very Good | 0–50 | 95.65 | 2.13 | 86.23 | 1.91 |
| 2. Good | 50–100 | 3586.92 | 79.76 | 1120.58 | 24.86 |
| 3. Moderate | 100–150 | 633.61 | 14.09 | 2428.58 | 53.88 |
| 4. Poor | 150–200 | 152.16 | 3.38 | 552.41 | 12.26 |
| 5. Very Poor | 200–250 | 7.37 | 0.16 | 209.65 | 4.65 |
| 6. Unfit | >250 | 21.64 | 0.48 | 110.11 | 2.44 |

Figure 8c,d indicate that the groundwater is categorized as unfit for drinking in the Vaiyampatti block during the pre-monsoon season and in the Vaiyampatti, Marungapuri, and Tiruverumbur blocks during the post-monsoon season. Therefore, according to the Rank Sum-based GQI maps, the groundwater quality is predicted to be unsuitable for drinking in southern parts of the study area. The disposal of sewage wastes containing treated wastewater, effluents from septic tanks, livestock wastes from confined animal facilities, and municipal wastes increase the concentration of TDS, $Cl^-$, TH, and $Na^+$ in the groundwater of the southern parts of the study area. On the other hand, the northern part is found to have a suitable quality of groundwater since it is the mountainous region with the predominant land area covered by forests and has the least human interventions in the land use/land cover condition, thereby leading to acceptable groundwater quality in this region.

### 3.3.3. GQI Maps Based on Analytic Hierarchy Process (AHP) Model

Using Saaty's AHP scale, weights were assigned to the ten themes (concentration maps of groundwater quality parameters) and their features (classes of individual concentration maps) based on the relative influence of the parameters on drinking water quality. Thereafter, pair-wise comparison matrices were prepared to compute the normalized weights of the ten themes and their features. As an illustration, the normalized weights obtained from the pair-wise comparison matrix prepared for the ten thematic layers are shown in Table 5. The weights assigned to the feature classes of each thematic layer and their normalized weights are presented in Table 6 for the pre-monsoon season and in Table 7 for the post-monsoon season. The value of the Consistency Ratio (CR) for the weights assigned to the themes was 3.6% and it ranges from 1.1% to 8.4% for the features of individual themes; these values of CR are well within the minimum threshold value of 10%. This indicates that the weights assigned to the themes and the respective feature classes are consistent. The thematic layers and their feature classes were then integrated with ArcGIS to obtain the seasonal GQI values.

**Table 5.** Pair-wise comparison matrix and normalized weights for the themes.

| Themes | $Mg^{2+}$ | $F^-$ | TDS | $Ca^{2+}$ | $Na^+$ | $SO_4^{2-}$ | $K^+$ | $Cl^-$ | $NO_3^-$-N | TH | Normalized Weights |
|---|---|---|---|---|---|---|---|---|---|---|---|
| $Mg^{2+}$ | 1 | 1/8 | 1/4 | 1/2 | 1/6 | 1/3 | 1/2 | 1/7 | 1/9 | 1/5 | 0.017 |
| $F^-$ | 8 | 1 | 5 | 7 | 3 | 6 | 7 | 2 | 1/2 | 4 | 0.215 |
| TDS | 4 | 1/5 | 1 | 3 | 1/3 | 2 | 3 | 1/4 | 1/6 | 1/2 | 0.052 |
| $Ca^{2+}$ | 2 | 1/7 | 1/3 | 1 | 1/5 | 1/2 | 1 | 1/6 | 1/8 | 1/4 | 0.024 |
| $Na^+$ | 6 | 1/3 | 3 | 5 | 1 | 4 | 5 | 1/2 | 1/4 | 2 | 0.109 |
| $SO_4^{2-}$ | 3 | 1/6 | 1/2 | 2 | 1/4 | 1 | 2 | 1/5 | 1/7 | 1/3 | 0.036 |
| $K^+$ | 2 | 1/7 | 1/3 | 1 | 1/5 | 1/2 | 1 | 1/6 | 1/8 | 1/4 | 0.024 |
| $Cl^-$ | 7 | 1/2 | 4 | 6 | 2 | 5 | 6 | 1 | 1/3 | 3 | 0.154 |
| $NO_3^-$-N | 9 | 2 | 6 | 8 | 4 | 7 | 8 | 3 | 1 | 5 | 0.292 |
| TH | 5 | 1/4 | 2 | 4 | 1/2 | 3 | 4 | 1/3 | 1/5 | 1 | 0.076 |
| Column Total | | | | | | | | | | | 1 |

**Table 6.** Weight assessment for groundwater quality parameters using AHP method for the **pre-monsoon season**.

| Sl. No. | Theme | Theme Weight | $W_i$ | Feature Class | Groundwater Prospect | Feature Weight | $w_j$ |
|---|---|---|---|---|---|---|---|
| 1 | $Mg^{2+}$ | 1 | 0.017 | <50 | Desirable Limit | 1 | 0.136 |
| | | | | 50–100 | Desirable Limit | 2 | 0.238 |
| | | | | 100–200 | Desirable Limit | 4 | 0.625 |
| 2 | $F^-$ | 8 | 0.215 | <0.6 | Very Less | 4 | 0.258 |
| | | | | 0.6–1.5 | Desirable Limit | 1 | 0.069 |
| | | | | 1.5–1.7 | Maximum Permissible Limit | 2 | 0.11 |
| | | | | >1.7 | Not Suitable | 6 | 0.562 |
| 3 | TDS | 4 | 0.052 | <500 | Desirable Limit | 1 | 0.081 |
| | | | | 500–1500 | Maximum Permissible Limit | 3 | 0.188 |
| | | | | >1500 | Not Suitable | 7 | 0.731 |
| 4 | $Ca^{2+}$ | 2 | 0.024 | <30 | Desirable Limit | 1 | 0.095 |
| | | | | 30–50 | Desirable Limit | 2 | 0.16 |
| | | | | 50–70 | Desirable Limit | 3 | 0.278 |
| | | | | >70 | Desirable Limit | 4 | 0.467 |
| 5 | $Na^+$ | 6 | 0.109 | <200 | Desirable Limit | 1 | 0.072 |
| | | | | 200–500 | Not Suitable | 5 | 0.279 |
| | | | | >500 | Not Suitable | 7 | 0.649 |
| 6 | $SO_4^{2-}$ | 3 | 0.036 | <75 | Desirable Limit | 1 | 0.069 |
| | | | | 75–100 | Desirable Limit | 2 | 0.11 |
| | | | | 100–250 | Desirable Limit | 4 | 0.258 |
| | | | | >250 | Not Suitable | 6 | 0.562 |
| 7 | $K^+$ | 2 | 0.24 | <10 | Desirable Limit | 2 | 0.105 |
| | | | | 10–50 | Not Suitable | 4 | 0.258 |
| | | | | >50 | Not Suitable | 6 | 0.637 |
| 8 | $Cl^-$ | 7 | 0.154 | <200 | Desirable Limit | 2 | 0.075 |
| | | | | 200–600 | Maximum Permissible Limit | 5 | 0.229 |
| | | | | >600 | Not Suitable | 8 | 0.696 |
| 9 | $NO_3^-$-N | 9 | 0.292 | <10 | Desirable Limit | 2 | 0.045 |
| | | | | 10–50 | Not Suitable | 6 | 0.156 |
| | | | | 50–100 | Not Suitable | 7 | 0.249 |
| | | | | >100 | Not Suitable | 9 | 0.55 |
| 10 | TH | 5 | 0.076 | 75–150 | Moderately Hard | 3 | 0.073 |
| | | | | 150–300 | Hard | 5 | 0.17 |
| | | | | 300–500 | Very Hard | 7 | 0.285 |
| | | | | 500–1000 | Very Hard | 8 | 0.472 |

**Table 7.** Weight assessment for groundwater quality parameters using AHP model for the **post-monsoon season**.

| Sl. No. | Theme | Theme Weight | $W_i$ | Feature Class | Groundwater Prospect | Feature Weight | $w_j$ |
|---|---|---|---|---|---|---|---|
| 1 | $Mg^{2+}$ | 1 | 0.017 | <50 | Desirable Limit | 1 | 0.046 |
| | | | | 50–100 | Desirable Limit | 2 | 0.069 |
| | | | | 100–200 | Desirable Limit | 4 | 0.154 |
| | | | | 200–300 | Desirable Limit | 5 | 0.238 |
| | | | | >300 | Not Suitable | 7 | 0.493 |
| 2 | $F^-$ | 8 | 0.215 | <0.6 | Very Less | 4 | 0.258 |
| | | | | 0.6–1.5 | Desirable Limit | 1 | 0.069 |
| | | | | 1.5–1.7 | Maximum Permissible Limit | 2 | 0.11 |
| | | | | >1.7 | Not Suitable | 6 | 0.562 |
| 3 | TDS | 4 | 0.052 | <500 | Desirable Limit | 1 | 0.081 |
| | | | | 500–1500 | Maximum Permissible Limit | 3 | 0.188 |
| | | | | >1500 | Not Suitable | 7 | 0.731 |
| 4 | $Ca^{2+}$ | 2 | 0.024 | <30 | Desirable Limit | 1 | 0.095 |
| | | | | 30–50 | Desirable Limit | 2 | 0.16 |
| | | | | 50–70 | Desirable Limit | 3 | 0.278 |
| | | | | >70 | Desirable Limit | 4 | 0.467 |
| 5 | $Na^+$ | 6 | 0.109 | <200 | Desirable Limit | 1 | 0.072 |
| | | | | 200–500 | Not Suitable | 5 | 0.279 |
| | | | | >500 | Not Suitable | 7 | 0.649 |
| 6 | $SO_4^{2-}$ | 3 | 0.036 | <75 | Desirable Limit | 1 | 0.136 |
| | | | | 75–100 | Desirable Limit | 2 | 0.238 |
| | | | | 100–250 | Desirable Limit | 4 | 0.625 |
| 7 | $K^+$ | 2 | 0.24 | <10 | Desirable Limit | 2 | 0.105 |
| | | | | 10–50 | Not Suitable | 4 | 0.258 |
| | | | | >50 | Not Suitable | 6 | 0.637 |
| 8 | $Cl^-$ | 7 | 0.154 | <200 | Desirable Limit | 2 | 0.075 |
| | | | | 200–600 | Maximum Permissible Limit | 5 | 0.229 |
| | | | | >600 | Not Suitable | 8 | 0.696 |
| 9 | $NO_3^--N$ | 9 | 0.292 | <10 | Desirable Limit | 2 | 0.045 |
| | | | | 10–50 | Not Suitable | 6 | 0.156 |
| | | | | 50–100 | Not Suitable | 7 | 0.249 |
| | | | | >100 | Not Suitable | 9 | 0.55 |
| 10 | TH | 5 | 0.076 | 150–300 | Hard | 5 | 0.061 |
| | | | | 300–500 | Very Hard | 7 | 0.133 |
| | | | | 500–1000 | Very Hard | 8 | 0.311 |
| | | | | >1000 | Very Hard | 9 | 0.495 |

The calculated GQI values were divided into six classes similar to the previous models to generate the AHP-based GQI maps for the pre-monsoon and post-monsoon seasons (Table 8). Figure 9a reveals that 5.7% and 44% of the study area falls in the 'Very good' and 'Good' zones, respectively, in the pre-monsoon season covering the northern and central parts of the study area. The 'Very Poor' groundwater class is found to be prominent in the Lalgudi block along with some small areas found in the Vaiyampatti, Tiruverumbur, Thottiyam, Manapparai, and Marungapuri blocks of the study area. Owing to its lower elevation, a predominant region under the Vaiyampatti block lying in the southwestern part has an 'Unfit' groundwater quality. In addition to this, agriculture is the predominant land use in this part of the study area, and the excessive usage of chemical fertilizers in the fields adds to the concentrations of TDS, $Cl^-$, $Mg^{2+}$, $NO_3^--N$, and $K^+$ in the groundwater

of this region. An excessive application of lime to the soil in agriculture-dominated areas is a major source for $Mg^{2+}$ concentration in the groundwater. The remaining part of the study area falls in the 'Moderate' (32.5%) to 'Poor' (12.5%) groundwater quality zones during the pre-monsoon season. In contrast to the pre-monsoon season, a major portion of the study area (80.97%) lies in the 'Good' groundwater quality class followed by the 'Moderate' (13.03%) and 'Poor' (3.14%) classes during the post-monsoon season [Figure 9b]. About 0.28% and 0.11% of the study area falls in the 'Very Poor' and 'Unfit' classes, respectively, encompassing the Tiruverumbur, Marungapuri, and Vaiyampatti blocks.

**Table 8.** Areal extent of groundwater quality zones predicted by the **AHP model**.

| Groundwater Quality Zone | GQI Range | Pre-Monsoon Season | | Post-Monsoon Season | |
|---|---|---|---|---|---|
| | | Area (km$^2$) | Area (%) | Area (km$^2$) | Area (%) |
| 1. Very Good | 0–50 | 257.01 | 5.71 | 110.66 | 2.46 |
| 2. Good | 50–100 | 1981.78 | 44.00 | 3648.99 | 80.97 |
| 3. Moderate | 100–150 | 1462.66 | 32.48 | 587.41 | 13.03 |
| 4. Poor | 150–200 | 563.88 | 12.52 | 141.50 | 3.14 |
| 5. Very Poor | 200–250 | 139.53 | 3.10 | 12.82 | 0.28 |
| 6. Unfit | >250 | 98.94 | 2.20 | 5.05 | 0.11 |

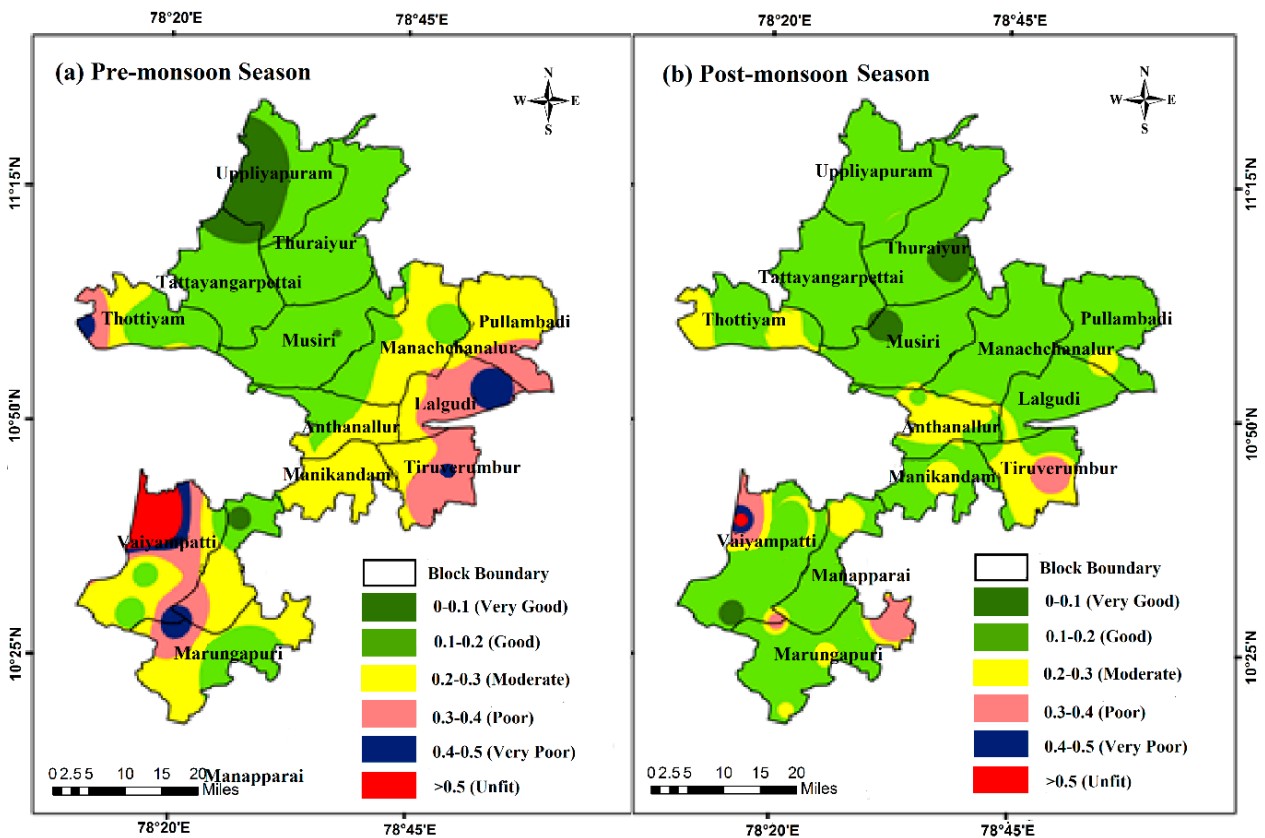

**Figure 9.** (**a**,**b**). GQI maps of the study area based on the **AHP model**; (**a**) Pre-monsoon Season; (**b**) Post-monsoon Season.

### 3.4. Comparison of the GQI Maps and Validation

A comparative evaluation of the GQI maps predicted by the three GIS-bases MCDA models is presented in Figure 10a,b. It can be seen from Figure 10a,b that in the case of the Unit Weight model, there is a considerable increase in the area occupied by the

'Very Good', 'Moderate', 'Poor', and 'Very Poor' groundwater quality zones during the post-monsoon season compared to the pre-monsoon season. However, the area under the 'Good' and 'Unfit' groundwater quality zones decreases in the post-monsoon season as per the Unit Weight model. The GQI maps predicted by the Rank Sum method depict an increase in the areal extent of all groundwater quality zones in the post-monsoon season compared to the pre-monsoon season expect for the 'Very Good' and 'Good' zones. On the other hand, the AHP model predicts a considerable increase in the area occupied by the 'Good' groundwater quality zones in the post-monsoon season than the pre-monsoon season, while there is a decrease in the area under all other zones. Despite these variations, the groundwater quality zones predicted by the three GIS-based models exhibit a similar pattern depicting a relatively poorer groundwater quality in the post-monsoon season than the pre-monsoon season. This reveals that the groundwater in the study area does not experience a dilution effect due to the monsoon rainfall. Moreover, the GQI values estimated by the Rank Sum model and AHP model are higher than the Unit Weight model. This indicates that the Unit Weight model is less sensitive to the variation in the concentrations of groundwater quality parameters than the other two models.

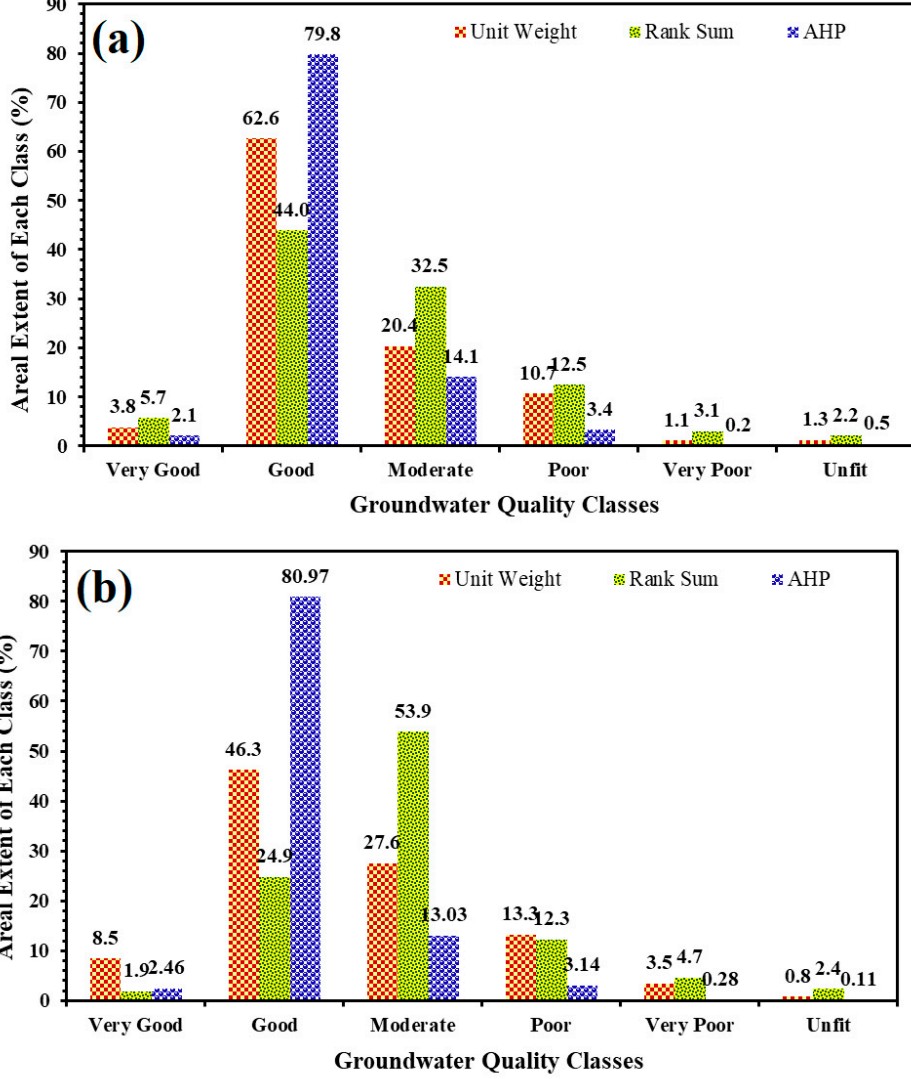

**Figure 10.** (**a**,**b**). Comparison of areas under each groundwater quality zone predicted by the three models: (**a**) Pre-monsoon Season; (**b**) Post-monsoon Season.

Furthermore, the Relative Operating Characteristics (ROC) curve technique was applied to validate the results obtained by the three models. Two sets of ROC curves were

prepared for each of the three models, one for the pre-monsoon season [Figure 11a] and the other for the post-monsoon [Figure 11b] season. The prediction accuracy of individual GQI maps is characterized by the area under the ROC curve. Figure 11a reveals that the prediction accuracy of the Unit Weight model and Rank Sum model lies in the 'average' category with values 0.68 and 0.64, respectively, in the pre-monsoon season. From this, it can be inferred that the Unit Weight model performs slightly better than the Rank Sum model in assessing the groundwater quality in the study area. In contrast, the AUC for the AHP model (0.71) is found to fall in the 'good' (0.7–0.8) class, indicating the best performance of the AHP model. The validation results for the post-monsoon season are shown in Figure 11b which suggests a better performance of the Rank Sum model (AUC = 0.68) than the Unit Weight model (AUC = 0.63). However, the AHP model is found to have 84.9% of the area under the ROC curve, thereby outperforming the other two models in predicting groundwater quality in the study area. Thus, it is concluded that the AHP model has very good capability in predicting groundwater quality irrespective of the season.

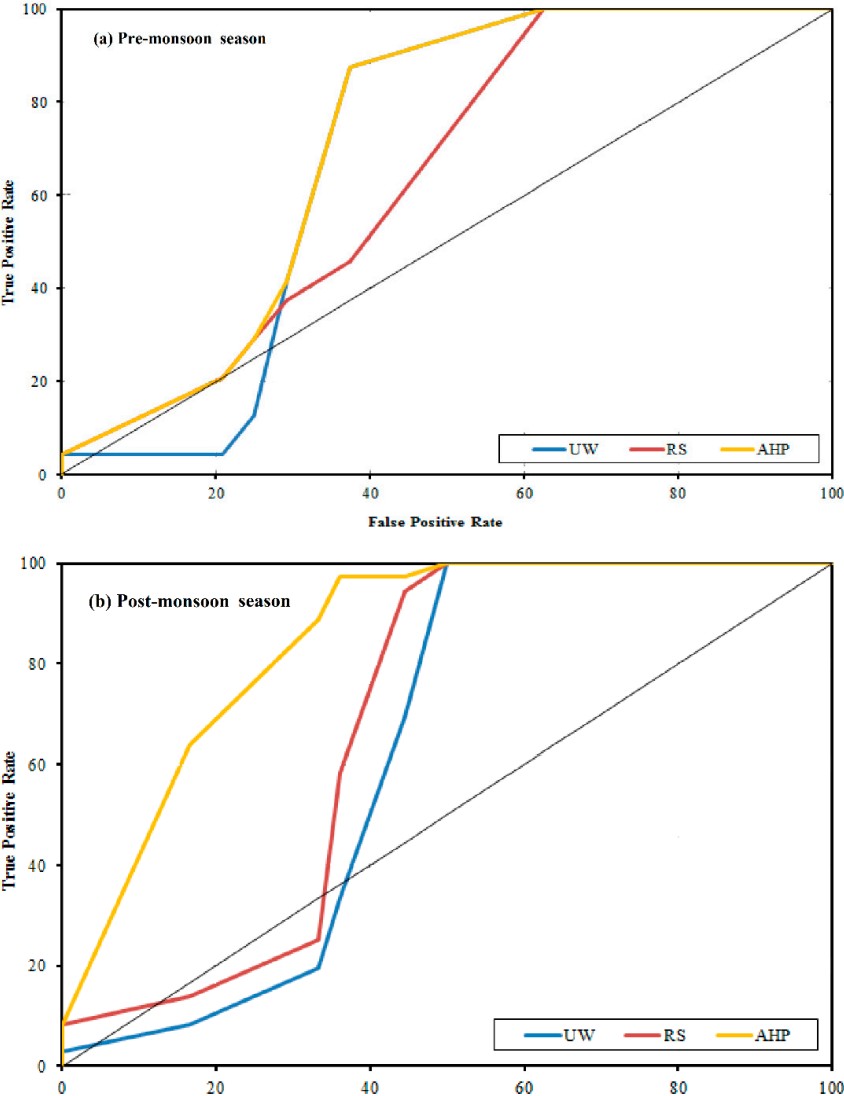

**Figure 11.** (**a**,**b**) ROC curves for the GQI maps predicted by the GIS-based MCDA models: (**a**) Pre-monsoon season; (**b**) Post-monsoon season.

## 4. Conclusions

In this study, for the first time, the effectiveness of three GIS-based Multi-Criteria Decision Analysis (MCDA) models, namely, the Unit Weight model, the Rank Sum model,

and the Analytic Hierarchy Process (AHP) model, were explored in predicting the suitability of groundwater as a potable water supply. The comparative analysis of these models was demonstrated with the help of a case study in Tiruchirappalli District, Tamil Nadu, India. For this, the concentration data of ten significant groundwater quality parameters ($Cl^-$, TDS, TH, $F^-$, $NO_3^-$-N, $Na^+$, $Mg^{2+}$, $Ca^{2+}$, $K^+$, and $SO_4^{2-}$) from 37 observation wells tapping the unconfined aquifers of the study area were considered. Initially, the preliminary analysis was performed using graphical and statistical techniques. Thereafter, three GIS-based GQI maps were developed for pre-monsoon and post-monsoon seasons using the significant groundwater quality parameters. The generated GQI maps were validated using a novel approach of the Relative Operating Characteristic (ROC) curve technique.

The results of the groundwater quality assessments using the graphical and statistical techniques revealed that the $Cl^-$ concentration exhibits a high temporal variation in the Uppliyapuram block during the pre-monsoon season (CV = 104.6%), while the remaining parameters exhibit low or moderate temporal variability. Furthermore, the results of the trend analysis indicate a significant increasing trend in the TDS, $Cl^-$, TH, $Na^+$, $Mg^{2+}$, and $K^+$ concentrations in groundwater during both seasons. The generated seasonal GQI maps reveal 'Very Good' to 'Moderate' groundwater quality in the northern, eastern, and central parts of the study area. Contradictory to this, the southern part of the study area has 'Poor' to 'Very Poor' groundwater quality, making it unsuitable for drinking. The validation of the generated GQI maps reveals that the AHP model outperforms with prediction accuracies of 71.4% in the pre-monsoon season and about 85% in the post-monsoon season in assessing groundwater quality. However, the performances of the Unit Weight model and Rank Sum model were found to be average with the prediction accuracies varying from 68% to 63% and 64% to 68%, respectively.

Based on the results of this study, it can be concluded that GIS-based MCDA models are effective in assessing the quality of groundwater in hard-rock aquifer systems. The AHP model is more efficient and hence it is recommended for groundwater quality evaluations in other regions of India. This study also reveals that the contamination of groundwater from the application of fertilizers and pesticides in the study region must be restricted since it is much more cost-effective to prevent the contamination of groundwater rather than the remediation of polluted groundwater resources. This study is very useful for policymakers to adopt viable groundwater management strategies for sustainable utilization of this vital resource. The methodological framework adopted in this study can be easily replicated in other regions of the world for groundwater quality evaluation irrespective of their hydrogeologic or climatic settings.

**Author Contributions:** M.A.J.: Formal analysis, Methodology, Visualization, Resources, Investigation, Validation, Writing—review & editing, M.K.J.: Project administration, Supervision, Data curation, Conceptualization, Writing—review & editing, Funding acquisition. A.K.: Writing—original draft, Formal analysis, Validation. All authors have read and agreed to the published version of the manuscript.

**Funding:** This research received no external funding.

**Institutional Review Board Statement:** Not applicable.

**Informed Consent Statement:** Not applicable.

**Data Availability Statement:** Not applicable.

**Acknowledgments:** The authors are very grateful to the Survey of India, Institute for Water Studies, and Chennai, Government of Tamil Nadu, India, for providing the data required to carry out the study. They would also like to express their sincere thanks and gratitude to the Ministry of Human Resource Development (MHRD), New Delhi, India, for the financial assistance received. The constructive comments provided by the two anonymous reviewers and the editor are also sincerely acknowledged.

**Conflicts of Interest:** The authors declare no conflict of interest.

# Appendix A

**Table A1.** Descriptive statistics of the groundwater quality parameters in the **pre-monsoon** season.

| Blocks | Statistics | TDS (mg/L) | Cl⁻ (mg/L) | TH (mg/L) | F⁻ (mg/L) | NO₃⁻-N (mg/L) | Na⁺ (mg/L) | Ca²⁺ (mg/L) | Mg²⁺ (mg/L) | K⁺ (mg/L) | SO₄²⁻ (mg/L) |
|---|---|---|---|---|---|---|---|---|---|---|---|
| 1. Lalgudi | Mean | 1292.4 | 484.4 | 542.0 | 0.4 | 18.6 | 277.9 | 87.8 | 74.5 | 74.1 | 43.8 |
| | SD | 342.4 | 184.5 | 183.0 | 0.2 | 3.3 | 25.8 | 13.0 | 8.7 | 19.9 | 8.1 |
| | CV | 0.265 | 0.381 | 0.338 | 0.500 | 0.177 | 0.093 | 0.148 | 0.117 | 0.269 | 0.185 |
| | Min | 797.5 | 282.0 | 265.0 | 0.1 | 28.3 | 118.5 | 41.0 | 32.5 | 0.0 | 33.0 |
| | Max | 2005.0 | 879.5 | 835.0 | 0.7 | 99.8 | 399.0 | 148.0 | 135.0 | 205.5 | 76.0 |
| | Trend | Increasing | Increasing ** | Increasing | Decreasing | Increasing | Increasing | Decreasing | Increasing | Increasing * | Increasing |
| 2. Manapparai | Mean | 1425.0 | 513.6 | 265.7 | 0.6 | 3.9 | 94.8 | 34.7 | 43.5 | 5.7 | 50.1 |
| | SD | 519.9 | 190.5 | 93.4 | 0.2 | 0.8 | 27.8 | 5.5 | 7.0 | 2.4 | 2.0 |
| | CV | 0.365 | 0.371 | 0.352 | 0.333 | 0.205 | 0.294 | 0.157 | 0.161 | 0.428 | 0.039 |
| | Min | 601.5 | 151.0 | 182.5 | 0.4 | 1.3 | 27.5 | 22.0 | 26.2 | 0.1 | 44.7 |
| | Max | 2176.0 | 753.5 | 470.0 | 1.0 | 7.0 | 210.5 | 57.0 | 79.6 | 19.5 | 56.7 |
| | Trend | Decreasing | Increasing | Increasing | Decreasing | Increasing | Increasing | Increasing | Increasing | Increasing | Decreasing |
| 3. Man-achchanallur | Mean | 723.8 | 217.4 | 399.2 | 0.6 | 23.4 | 109.3 | 53.0 | 64.8 | 13.8 | 58.7 |
| | SD | 191.4 | 99.7 | 80.0 | 0.2 | 2.8 | 7.9 | 3.9 | 3.8 | 4.2 | 10.6 |
| | CV | 0.264 | 0.459 | 0.200 | 0.333 | 0.120 | 0.072 | 0.074 | 0.058 | 0.306 | 0.181 |
| | Min | 213.5 | 70.8 | 276.3 | 0.3 | 3.3 | 29.8 | 30.5 | 42.8 | 0.0 | 38.5 |
| | Max | 1083.3 | 518.5 | 555.0 | 1.0 | 64.8 | 191.3 | 100.5 | 106.6 | 96.3 | 95.5 |
| | Trend | Increasing | Increasing | Increasing | Decreasing | Decreasing | Increasing | Decreasing | Increasing | Decreasing | Decreasing |
| 4. Marungapuri | Mean | 1032.8 | 323.3 | 481.9 | 0.7 | 7.8 | 261.9 | 51.5 | 99.4 | 12.6 | 52.8 |
| | SD | 506.1 | 178.3 | 139.3 | 0.2 | 1.2 | 43.9 | 9.3 | 17.7 | 2.8 | 5.9 |
| | CV | 0.490 | 0.552 | 0.289 | 0.286 | 0.154 | 0.168 | 0.181 | 0.178 | 0.226 | 0.111 |
| | Min | 366.5 | 59.8 | 255.0 | 0.3 | 0.5 | 165.0 | 15.3 | 35.3 | 0.7 | 16.0 |
| | Max | 1728.8 | 549.3 | 691.7 | 0.9 | 28.0 | 481.3 | 118.7 | 250.0 | 43.7 | 89.0 |
| | Trend | Decreasing ** | Decreasing * | Decreasing * | Decreasing | Decreasing | Decreasing ** | Increasing | Increasing | Increasing | Increasing |
| 5. Musiri | Mean | 661.6 | 190.0 | 357.8 | 0.5 | 17.5 | 108.8 | 52.0 | 50.9 | 11.0 | 76.0 |
| | SD | 196.2 | 110.4 | 116.3 | 0.1 | 2.5 | 6.9 | 3.3 | 4.3 | 1.5 | 13.5 |
| | CV | 0.297 | 0.581 | 0.325 | 0.200 | 0.143 | 0.063 | 0.064 | 0.084 | 0.138 | 0.178 |
| | Min | 282.5 | 76.0 | 237.5 | 0.3 | 1.5 | 32.0 | 23.0 | 21.5 | 0.0 | 19.0 |
| | Max | 1186.0 | 629.0 | 640.0 | 0.9 | 41.0 | 200.0 | 110.0 | 141.5 | 39.5 | 169.0 |
| | Trend | Decreasing | Increasing | Increasing | Decreasing | Decreasing * | Decreasing | Decreasing | Increasing | Increasing * | Decreasing |

**Table A1.** *Cont.*

| Blocks | Statistics | TDS (mg/L) | Cl⁻ (mg/L) | TH (mg/L) | F⁻ (mg/L) | NO₃⁻-N (mg/L) | Na⁺ (mg/L) | Ca²⁺ (mg/L) | Mg²⁺ (mg/L) | K⁺ (mg/L) | SO₄²⁻ (mg/L) |
|---|---|---|---|---|---|---|---|---|---|---|---|
| 6. Pullambadi | Mean | 696.3 | 190.0 | 383.4 | 0.5 | 20.8 | 86.7 | 64.3 | 55.2 | 9.6 | 66.0 |
| | SD | 271.2 | 121.0 | 126.9 | 0.1 | 4.4 | 17.7 | 7.4 | 5.1 | 3.2 | 8.9 |
| | CV | 0.389 | 0.637 | 0.331 | 0.200 | 0.212 | 0.205 | 0.115 | 0.093 | 0.334 | 0.135 |
| | Min | 335.5 | 37.0 | 165.0 | 0.4 | 10.0 | 8.0 | 31.0 | 17.0 | 0.0 | 27.0 |
| | Max | 1337.0 | 463.0 | 665.0 | 0.6 | 50.0 | 313.0 | 164.0 | 104.0 | 52.5 | 132.0 |
| | Trend | Increasing * | Increasing * | Increasing ** | Decreasing | Increasing | Increasing * | Increasing | Increasing ** | Increasing | Increasing |
| 7. Thottiyam | Mean | 792.3 | 199.8 | 293.7 | 0.7 | 12.7 | 179.0 | 43.6 | 44.9 | 5.7 | 37.6 |
| | SD | 190.2 | 74.9 | 75.7 | 0.3 | 2.1 | 13.8 | 3.6 | 3.8 | 1.3 | 4.7 |
| | CV | 0.240 | 0.375 | 0.258 | 0.429 | 0.165 | 0.077 | 0.083 | 0.084 | 0.223 | 0.126 |
| | Min | 393.0 | 75.7 | 193.3 | 0.4 | 3.3 | 61.3 | 24.7 | 25.0 | 0.0 | 18.0 |
| | Max | 1081.3 | 332.0 | 480.0 | 1.5 | 24.0 | 287.0 | 78.7 | 88.0 | 18.3 | 70.0 |
| | Trend | Increasing | Increasing * | Increasing | Increasing | Decreasing ** | Increasing ** | Decreasing | Increasing | Increasing * | Decreasing |
| 8. Tiruverumbur | Mean | 806.8 | 261.7 | 315.2 | 0.4 | 8.2 | 158.9 | 56.9 | 41.7 | 37.7 | 44.9 |
| | SD | 186.3 | 101.3 | 96.7 | 0.1 | 1.5 | 15.7 | 10.4 | 5.5 | 6.6 | 3.7 |
| | CV | 0.231 | 0.387 | 0.307 | 0.250 | 0.183 | 0.099 | 0.183 | 0.133 | 0.174 | 0.081 |
| | Min | 365.5 | 89.0 | 165.0 | 0.2 | 1.0 | 52.5 | 19.0 | 20.4 | 8.0 | 25.0 |
| | Max | 1062.5 | 439.5 | 470.0 | 0.5 | 16.0 | 219.5 | 152.0 | 82.0 | 69.5 | 76.0 |
| | Trend | Increasing | Increasing | Increasing | Decreasing | Increasing | Increasing | Decreasing | Increasing ** | Decreasing ** | Decreasing |
| 9. Uppliyapu-ram | Mean | 410.7 | 92.8 | 266.7 | 0.5 | 7.5 | 50.9 | 32.3 | 45.3 | 11.3 | 70.4 |
| | SD | 226.5 | 97.1 | 111.0 | 0.2 | 2.7 | 9.0 | 1.9 | 4.6 | 1.9 | 13.9 |
| | CV | 0.551 | 1.046 | 0.416 | 0.400 | 0.360 | 0.176 | 0.059 | 0.101 | 0.169 | 0.197 |
| | Min | 199.5 | 23.0 | 50.0 | 0.2 | 1.0 | 12.5 | 12.0 | 5.0 | 0.0 | 27.3 |
| | Max | 1455.0 | 553.5 | 722.5 | 0.9 | 57.0 | 253.0 | 61.0 | 148.0 | 39.0 | 128.0 |
| | Trend | Increasing | Increasing | Decreasing | Increasing | Decreasing | Increasing | Decreasing | Decreasing | Increasing | Decreasing |
| 10. Vaiyampatti | Mean | 761.7 | 237.4 | 389.1 | 0.9 | 9.2 | 132.6 | 51.9 | 61.2 | 2.6 | 29.8 |
| | SD | 126.4 | 61.8 | 86.2 | 0.2 | 2.2 | 10.4 | 5.4 | 4.1 | 1.3 | 2.5 |
| | CV | 0.166 | 0.260 | 0.222 | 0.222 | 0.239 | 0.079 | 0.104 | 0.067 | 0.505 | 0.082 |
| | Min | 547.6 | 133.2 | 256.3 | 0.6 | 1.5 | 93.2 | 30.4 | 44.8 | 0.0 | 24.0 |
| | Max | 998.0 | 392.2 | 547.5 | 1.2 | 32.0 | 207.0 | 74.6 | 81.0 | 9.0 | 36.0 |
| | Trend | Increasing | Increasing | Increasing | Increasing | Decreasing | Increasing | Decreasing | Increasing | Increasing * | Decreasing |

Note: SD = Standard Deviation; CV = Coefficient of Variation (Red Text indicates CV < 10%); * Significant at 1% significance level; ** Significant at 5% significance level (Red Text indicates Significant Trend).

**Table A2.** Descriptive statistics of groundwater quality parameters in the **post-monsoon** season.

| Blocks² | Statistics | TDS (mg/L) | Cl⁻ (mg/L) | TH (mg/L) | F⁻ (mg/L) | NO₃⁻-N (mg/L) | Na⁺ (mg/L) | Ca²⁺ (mg/L) | Mg²⁺ (mg/L) | K⁺ (mg/L) | SO₄²⁻ (mg/L) |
|---------|-----------|-----------|-----------|-----------|-----------|--------------|-----------|------------|------------|-----------|-------------|
| 1. Lalgudi | Mean | 1404.5 | 499.3 | 535.5 | 0.5 | 24.8 | 302.2 | 92.6 | 84.4 | 63.7 | 21.2 |
|  | SD | 565.8 | 200.8 | 190.4 | 0.2 | 5.4 | 28.7 | 9.8 | 12.1 | 23.1 | 4.0 |
|  | CV | 0.403 | 0.402 | 0.356 | 0.400 | 0.218 | 0.095 | 0.106 | 0.143 | 0.362 | 0.190 |
|  | Min | 628.0 | 212.5 | 292.5 | 0.3 | 5.0 | 138.5 | 66.0 | 41.5 | 0.0 | 32.4 |
|  | Max | 2944.0 | 929.0 | 950.0 | 0.9 | 64.5 | 621.0 | 144.0 | 128.0 | 172.0 | 62.4 |
|  | Trend | Decreasing | Decreasing | Increasing | Decreasing | Increasing | Decreasing | Increasing | Decreasing | Increasing | Decreasing |
| 2. Manapparai | Mean | 1038.6 | 402.2 | 394.6 | 0.5 | 13.5 | 359.8 | 46.1 | 67.9 | 11.4 | 55.6 |
|  | SD | 475.5 | 224.8 | 258.7 | 0.1 | 3.5 | 61.8 | 9.4 | 21.7 | 3.2 | 11.5 |
|  | CV | 0.458 | 0.559 | 0.656 | 0.200 | 0.259 | 0.172 | 0.204 | 0.319 | 0.283 | 0.208 |
|  | Min | 272.0 | 53.0 | 197.5 | 0.4 | 1.0 | 157.5 | 18.0 | 29.5 | 0.0 | 32.0 |
|  | Max | 1881.0 | 808.0 | 1000.0 | 0.7 | 25.5 | 607.0 | 80.0 | 194.5 | 22.5 | 95.0 |
|  | Trend | Decreasing ** | Decreasing ** | Decreasing | Decreasing | Increasing | Increasing | Decreasing | Decreasing | Increasing | Decreasing |
| 3. Man-achchanallur | Mean | 803.2 | 228.7 | 471.4 | 0.6 | 30.6 | 124.8 | 58.2 | 55.0 | 12.2 | 54.4 |
|  | SD | 214.2 | 105.7 | 171.4 | 0.1 | 4.7 | 19.6 | 9.7 | 3.7 | 1.3 | 5.6 |
|  | CV | 0.267 | 0.462 | 0.364 | 0.167 | 0.154 | 0.157 | 0.166 | 0.068 | 0.108 | 0.103 |
|  | Min | 607.8 | 119.5 | 235.0 | 0.4 | 4.0 | 67.5 | 22.5 | 38.0 | 9.0 | 39.0 |
|  | Max | 1404.0 | 509.1 | 920.0 | 0.8 | 68.7 | 188.8 | 85.5 | 64.5 | 17.3 | 74.0 |
|  | Trend | Increasing | Increasing | Decreasing | Decreasing | Increasing | Decreasing | Decreasing | Increasing | Increasing | Decreasing |
| 4. Marungapuri | Mean | 944.1 | 362.9 | 363.2 | 0.9 | 11.4 | 192.5 | 45.2 | 87.3 | 13.4 | 16.8 |
|  | SD | 344.6 | 139.8 | 140.7 | 0.3 | 2.9 | 36.0 | 4.5 | 14.6 | 2.7 | 3.3 |
|  | CV | 0.365 | 0.385 | 0.387 | 0.333 | 0.254 | 0.187 | 0.099 | 0.168 | 0.199 | 0.198 |
|  | Min | 562.3 | 170.3 | 132.5 | 0.6 | 5.3 | 92.3 | 30.5 | 42.8 | 0.0 | 35.0 |
|  | Max | 1741.8 | 761.5 | 710.0 | 1.4 | 29.5 | 364.0 | 62.0 | 159.8 | 22.5 | 52.5 |
|  | Trend | Increasing | Increasing | Increasing | Increasing | Increasing | Increasing | Decreasing | Increasing | Increasing | Increasing |
| 5. Musiri | Mean | 843.3 | 241.1 | 391.2 | 0.7 | 26.7 | 155.7 | 65.6 | 55.5 | 11.0 | 59.7 |
|  | SD | 385.5 | 124.5 | 118.8 | 0.2 | 6.6 | 23.2 | 6.3 | 4.5 | 3.4 | 12.2 |
|  | CV | 0.457 | 0.516 | 0.304 | 0.286 | 0.247 | 0.149 | 0.097 | 0.082 | 0.309 | 0.204 |
|  | Min | 498.0 | 133.0 | 215.0 | 0.5 | 3.5 | 66.0 | 29.0 | 30.0 | 0.0 | 32.0 |
|  | Max | 2281.5 | 730.0 | 760.0 | 1.0 | 95.5 | 444.0 | 112.0 | 116.5 | 60.5 | 108.0 |
|  | Trend | Increasing | Increasing | Increasing | Increasing | Decreasing | Increasing ** | Increasing | Increasing | Increasing ** | Decreasing |

**Table A2.** *Cont.*

| Blocks² | Statistics | TDS(mg/L) | Cl⁻ (mg/L) | TH (mg/L) | F⁻ (mg/L) | $NO_3^-$-N (mg/L) | $Na^+$ (mg/L) | $Ca^{2+}$ (mg/L) | $Mg^{2+}$ (mg/L) | $K^+$(mg/L) | $SO_4^{2-}$ (mg/L) |
|---|---|---|---|---|---|---|---|---|---|---|---|
| 6. Pullambadi | Mean | 550.1 | 137.8 | 303.8 | 0.2 | 16.3 | 82.4 | 54.4 | 41.6 | 6.1 | 40.8 |
| | SD | 145.1 | 59.7 | 77.9 | 0.1 | 2.5 | 8.7 | 5.3 | 3.1 | 1.4 | 4.4 |
| | CV | 0.264 | 0.433 | 0.256 | 0.500 | 0.153 | 0.106 | <span style="color:red">0.098</span> | <span style="color:red">0.073</span> | 0.230 | 0.107 |
| | Min | 346.0 | 58.5 | 170.0 | 0.1 | 7.0 | 24.0 | 23.0 | 15.5 | 0.0 | 29.5 |
| | Max | 789.0 | 244.5 | 445.0 | 0.4 | 27.0 | 150.5 | 98.0 | 60.0 | 20.0 | 53.0 |
| | Trend | Increasing | Increasing | Increasing | <span style="color:red">Decreasing **</span> | Increasing | Increasing | Decreasing | Increasing | <span style="color:red">Increasing *</span> | Decreasing |
| 7. Thottiyam | Mean | 961.1 | 259.7 | 358.5 | 0.5 | 26.3 | 208.2 | 55.8 | 52.6 | 23.4 | 57.3 |
| | SD | 296.7 | 126.3 | 127.5 | 0.3 | 3.7 | 19.0 | 6.4 | 5.5 | 13.6 | 9.6 |
| | CV | 0.309 | 0.486 | 0.356 | 0.600 | 0.141 | <span style="color:red">0.091</span> | 0.114 | 0.104 | 0.583 | 0.167 |
| | Min | 518.0 | 64.0 | 166.7 | 0.3 | 2.5 | 67.7 | 22.0 | 17.7 | 0.0 | 30.0 |
| | Max | 1586.0 | 533.0 | 665.0 | 1.1 | 66.0 | 383.7 | 128.0 | 98.3 | 263.0 | 83.0 |
| | Trend | <span style="color:red">Increasing *</span> | <span style="color:red">Increasing *</span> | <span style="color:red">Increasing **</span> | Decreasing | Increasing | Increasing | Increasing | <span style="color:red">Increasing *</span> | Increasing | Decreasing |
| 8. Tiruverumbur | Mean | 690.5 | 237.8 | 260.8 | 0.4 | 7.1 | 150.7 | 46.8 | 35.0 | 14.1 | 18.7 |
| | SD | 163.1 | 102.1 | 64.3 | 0.1 | 2.5 | 15.1 | 6.7 | 6.0 | 3.6 | 2.8 |
| | CV | 0.236 | 0.429 | 0.247 | 0.250 | 0.352 | 0.100 | 0.143 | 0.172 | 0.253 | 0.147 |
| | Min | 490.0 | 101.0 | 197.5 | 0.3 | 0.5 | 92.5 | 25.0 | 20.7 | 1.5 | 40.5 |
| | Max | 952.5 | 367.0 | 395.0 | 0.6 | 17.8 | 201.0 | 65.0 | 57.1 | 24.0 | 56.0 |
| | Trend | Increasing | Increasing | Increasing | <span style="color:red">Decreasing *</span> | Decreasing | Increasing | Decreasing | <span style="color:red">Increasing *</span> | Increasing | Decreasing |
| 9. Uppliyapuram | Mean | 428.3 | 80.7 | 289.2 | 0.6 | 16.9 | 47.1 | 36.3 | 47.4 | 11.5 | 49.9 |
| | SD | 167.8 | 64.9 | 103.6 | 0.2 | 3.2 | 7.4 | 4.0 | 5.8 | 3.3 | 7.8 |
| | CV | 0.392 | 0.804 | 0.358 | 0.333 | 0.189 | 0.157 | 0.111 | 0.122 | 0.290 | 0.157 |
| | Min | 233.5 | 21.0 | 162.5 | 0.3 | 2.5 | 10.5 | 12.0 | 29.0 | 0.0 | 31.0 |
| | Max | 758.5 | 289.0 | 585.0 | 0.8 | 39.0 | 123.0 | 70.0 | 123.5 | 41.5 | 71.0 |
| | Trend | Increasing | Decreasing | Increasing | Decreasing | Decreasing | Increasing | Increasing | Decreasing | Increasing | Decreasing |
| 10. Vaiyampatti | Mean | 896.9 | 293.9 | 418.4 | 0.8 | 16.9 | 165.1 | 50.4 | 72.0 | 15.2 | 51.3 |
| | SD | 344.0 | 149.8 | 129.1 | 0.1 | 5.1 | 18.2 | 5.6 | 7.0 | 4.0 | 8.7 |
| | CV | 0.384 | 0.510 | 0.309 | 0.125 | 0.302 | 0.110 | 0.112 | <span style="color:red">0.098</span> | 0.266 | 0.170 |
| | Min | 372.4 | 84.2 | 223.0 | 0.6 | 5.4 | 49.8 | 19.2 | 27.8 | 0.0 | 30.0 |
| | Max | 1809.4 | 733.8 | 663.0 | 1.0 | 50.6 | 381.2 | 111.6 | 149.4 | 54.6 | 82.0 |
| | Trend | Increasing | Increasing | Increasing | Increasing | Increasing | Increasing | <span style="color:red">Decreasing *</span> | Increasing | <span style="color:red">Increasing *</span> | Decreasing |

Note: SD = Standard Deviation; CV = Coefficient of Variation (Red Text indicates CV < 10%); * Significant at 1% significance level; ** Significant at 5% significance level (Red Text indicates Significant Trend).

**Table A3.** Correlation matrix of groundwater quality parameters during **pre-monsoon** season.

| | TDS | TH | Mg$^{2+}$ | Ca$^{2+}$ | Na$^+$ | K$^+$ | Cl$^-$ | SO$_4{}^{2-}$ | F$^-$ | NO$_3{}^-$-N |
|---|---|---|---|---|---|---|---|---|---|---|
| **TDS** | 1 | | | | | | | | | |
| **TH** | **0.93** | 1 | | | | | | | | |
| **Mg$^{2+}$** | −0.03 | 0.00 | 1 | | | | | | | |
| **Ca$^{2+}$** | −0.08 | −0.02 | **0.93** | 1 | | | | | | |
| **Na$^+$** | **0.86** | **0.67** | 0.19 | 0.09 | 1 | | | | | |
| **K$^+$** | 0.37 | 0.40 | 0.31 | 0.33 | 0.15 | 1 | | | | |
| **Cl$^-$** | **0.93** | **0.84** | 0.21 | 0.14 | **0.92** | 0.34 | 1 | | | |
| **SO$_4{}^{2-}$** | **0.77** | **0.79** | 0.32 | 0.17 | **0.62** | 0.45 | **0.75** | 1 | | |
| **F$^-$** | 0.14 | 0.24 | −0.14 | −0.26 | 0.09 | −0.37 | 0.03 | 0.24 | 1 | |
| **NO$_3{}^-$-N** | 0.48 | 0.539 | 0.03 | 0.10 | 0.48 | −0.09 | 0.45 | 0.25 | −0.10 | 1 |

Note: Bold values represent a significant correlation.

**Table A4.** Correlation matrix of groundwater quality parameters during **post-monsoon** season.

| | TDS | TH | Mg$^{2+}$ | Ca$^{2+}$ | Na$^+$ | K$^+$ | Cl$^-$ | SO$_4{}^{2-}$ | F$^-$ | NO$_3{}^-$-N |
|---|---|---|---|---|---|---|---|---|---|---|
| **TDS** | 1 | | | | | | | | | |
| **TH** | **0.90** | 1 | | | | | | | | |
| **Mg$^{2+}$** | **0.90** | **0.95** | 1 | | | | | | | |
| **Ca$^{2+}$** | **0.80** | **0.88** | **0.85** | 1 | | | | | | |
| **Na$^+$** | **0.83** | **0.52** | **0.52** | 0.44 | 1 | | | | | |
| **K$^+$** | 0.15 | 0.07 | 0.06 | 0.15 | 0.07 | 1 | | | | |
| **Cl$^-$** | **0.96** | **0.95** | **0.95** | **0.84** | **0.69** | 0.10 | 1 | | | |
| **SO$_4{}^{2-}$** | **0.63** | **0.52** | **0.53** | 0.38 | **0.57** | 0.10 | **0.58** | 1 | | |
| **F$^-$** | 0.41 | 0.22 | 0.26 | −0.03 | 0.45 | −0.23 | 0.28 | 0.20 | 1 | |
| **NO$_3{}^-$-N** | **0.83** | **0.79** | **0.79** | **0.69** | **0.60** | 0.01 | **0.84** | 0.42 | 0.27 | 1 |

Note: Bold values represent a significant correlation.

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
