# Peer review of "Assessing Multi-Criteria Decision Analysis Models for Predicting Groundwater Quality in a River Basin of South India"

_sustainability, doi:10.3390/su13126719_

Round 1

Reviewer 1 Report

The study applied geoinformatics and modeling for assessing groundwater quality in South India. The study is interesting, however needs some revisions.

Line 17….Elaborate GIS….Geographical Information System (GIS)

Line 20….Use superscript for the water quality indices

Line 27….Elaborate AHP….Analytic Hierarchy Process (AHP)

Line 43….Give comma after Andhra Pradesh

Line 61….Elaborate GIS….Geographical Information System (GIS)….as used for the first time.

Line 128….Write subtropical instead of Sub-tropical

The descriptive statistics of the groundwater-quality parameters in the pre- and post-monsoon seasons may be provided in the Supplementary data section.

Line 144….Elaborate EC and TDS in Line 144 instead of elaborating these in Line 147 & 148. Similarly, the names of all the cations and anions should also be described, where used first in the manuscript; e.g. [sodium (Na-), magnesium (Mg2+), etc.]

Line 145….What is redundancy K+?

Line 145….“from 37 observation wells”….Was the sampling georeferenced?

Line 168….Write seasons

Line 254….Write depicted instead of shown

Line 602….Give space in AHPmodel

Why pH and EC parameters were not included?

The study lacks discussion. Explain the changes obtained in water quality indices (concentration of cations and anions) during pre- and post-monsoon seasons.

The correlation matrix for the groundwater parameters may also be provided in the Supplementary data section.

Please follow the MDPI style for citations and references.

Reviewer 2 Report

The manuscript is sound interesting and authors have taken a great effort to introduce a novel approach to analyze the quality of the ground water.

The manuscript is well written and key points have been described sufficiently.  However, check the following points before publish.

  1. Improve the conclusion only highlighting the important points.
  2. Check the chemical formulas once again.

Round 2

Reviewer 1 Report

The author has successfully replied to my comment. The manuscript doesn't require further revision, I recommend accepting the manuscript in the present form. 

Author Response

The authors are thankful to the reviewer for accepting the manuscript
